# LENS: Learning to Navigate with Active Search for Partially Observable MAPF in Unknown Environments

**Di Yang**                                                                             *dyang06@wm.edu*
*College of William & Mary*

**Yuheng Li**                                                                            *yli95@wm.edu*
*College of William & Mary*

**Yanhai Xiong**                                                                  *yanhaixiong7@gmail.com*
*College of William & Mary*

**Reviewed on OpenReview:** *https://openreview.net/forum?id=DuNU6ZN5GG*

## Abstract

Classical Multi-Agent Path Finding (MAPF) solvers guarantee collision-free coordination but rely on perfect global knowledge, limiting their applicability in strictly unknown environments. Consequently, modern learning-based approaches face a dichotomy: decentralized reactive heuristics scale under partial observability but fail at structured deadlocks due to limited horizons and weak interaction inductive biases, while neural foundation models (e.g., MAPF-GPT) provide topological awareness but require pre-computed global heuristics and prohibitive training data.

We address Centralized Collaborative Partially Observable MAPF (PO-MAPF) by proposing LENS (**LE**arning to **N**avigate with active **S**earch), a hybrid architecture decoupling topological guidance from local collision avoidance. LENS employs a lightweight neural network that maps each agent's field of view (FOV) and goal direction into a dense local potential field to generate multi-step subpaths. Upon anticipating collisions, agents aggregate local observations into a shared belief, projecting neural proposal endpoints as waypoints. LENS then partitions anticipated conflicts into disjoint graphs and invokes localized Conflict-Based Search (L-CBS) over bounded spatio-temporal windows. This ensures strictly collision-free execution within a receding horizon, overcoming reactive myopia without the exponential overhead of global replanning.

Evaluations show that given full global knowledge, LENS approximates the solution quality of centralized oracles across most benchmarks and achieves out-of-distribution generalization comparable to 85M-parameter foundation models using $< 0.2\%$ of their training data. In strictly unknown environments with online mapping, LENS outperforms reactive baselines, improving success rates by $41.5\%$ in high-density mazes. By decoupling topological inference from collision resolution, LENS provides a scalable, data-efficient, and locally collision-free solution for autonomous navigation.

## 1 Introduction

Multi-Agent Path Finding (MAPF) serves as the algorithmic backbone for numerous large-scale robotic applications, ranging from automated warehousing (Li et al., 2021) to urban traffic management (Chen et al., 2024). While classical centralized algorithms, such as Conflict-Based Search (CBS) (Sharon et al., 2015; Boyarski et al., 2015) and LaCAM (Okumura, 2023), guarantee optimality or completeness(Surynek, 2010), they rely heavily on two assumptions: full observability of the environment and the availability of a perfect global heuristic(Ma et al., 2021; Mac et al., 2016; Ferguson et al., 2005; Varambally et al., 2022) (e.g.,

pre-computed BFS). In realistic deployment scenarios like search-and-rescue, these assumptions collapse, defining the rigorous setting of navigation in strictly unknown environments under Partial Observability (POMAPF), where agents lack both a prior map and a global view of dynamic peers(Liu et al., 2020b; Wang et al., 2020; Skrynnik et al., 2024b; Li et al., 2022b).

Recent MAPF scaling efforts present a dichotomy. *Decentralized reactive heuristics* (e.g., CS-PIBT (Veera-paneni et al., 2025; Okumura et al., 2022), LNS2-RL (Wang et al., 2025)) adapt well to partial observability but suffer from myopia and weak interaction inductive biases (Jain et al., 2026), making them susceptible to *structured deadlocks* (e.g., U-shaped traps or pairwise bottlenecks). Conversely, *neural foundation models* (e.g., MAPF-GPT (Andreychuk et al., 2025b)) provide topological foresight but rely on pre-computed global heuristics, which render them inapplicable in unknown environments and demand prohibitive training data.

In this paper, we bridge the gap between myopic, efficient heuristics and data-hungry, global learners by proposing **LENS** (*LEarning to Navigate with Active Search*). We argue that learning and search should play complementary(Veerapaneni et al., 2025), rather than competitive roles. LENS treats learning as a mechanism for Heuristic Approximation(Pohl, 1970; Bander & White, 2002), which infers dense local potential fields conditioned on global intent from partial observations(Hwang et al., 1992; Aggarwal & Kumar, 2020), while delegating the rigorous logic of collision avoidance to a specialized search routine. By inverting the standard paradigm, we utilize learning for global guidance and search for local resolution. As a result, LENS achieves topology-aware coordination under partial observability while maintaining local collision-free within bounded spatio-temporal windows, without requiring large-scale end-to-end policy learning.

Our main contributions are summarized as follows:

**Hierarchical Topo-metric Inference.** Unlike prior reactive policies (e.g., CS-PIBT, SCRIMP) that output 1-step discrete actions, or conventional hybrid frameworks (e.g., LNS2-RL) that rely on global search paired with local learned repair, LENS inverts this along two axes: search runs locally over anticipated conflict groups (not globally), and learning regresses a continuous potential field (not discrete repair actions). This dense prediction mitigates the myopia of 1-step reactive methods, enabling agents to recognize and navigate structural bottlenecks. We quantify this mitigation empirically: in **Long-Horizon Corridor** environments with high agent density, reactive policies (CS-PIBT) collapse to a 20.0% Success Rate due to severe topological livelocks, whereas LENS's multi-hop gradient maintains a 56.2% Success Rate (detailed in Appendix B).

**Active Search Injection via Cascade.** Second, we challenge the standard reactive shielding paradigm (e.g., CS-PIBT (Yukhnevich & Andreychuk, 2026; Veerapaneni et al., 2025; Okumura et al., 2022) by seamlessly coupling continuous neural heuristics with discrete combinatorial search. LENS incorporates a dynamic Cascade where the inferred potential field is utilized to project topology-aware local waypoints. Allowing us to bound an L-CBS to strictly route agents toward these dynamic subgoals, efficiently resolving structured deadlocks. This architectural inversion actively invokes search to resolve anticipated deadlocks before they occur, providing bounded collision-free coordination within localized spatio-temporal windows.

**Data-Efficient Generalization and Online Robustness.** Finally, with a decoupled evaluation protocol, we demonstrate that LENS systematically resolves the aforementioned dichotomy. In unknown environments requiring online collaborative mapping, it improves the success rate over reactive baselines by 41.5% in high-density topologies. Furthermore, under global knowledge settings, it achieves competitive out-of-distribution robustness comparable to foundation models with up to 85M parameters (e.g., MAPF-GPT), while utilizing less than 0.2% of their training data, establishing a highly resource-efficient paradigm for swarm navigation.

## 2 Related Work

### 2.1 Decentralized Heuristics and Reactive Policies

Learning-based extensions like CS-PIBT (Veerapaneni et al., 2025) combine a lightweight policy with a collision shield (PIBT)(Okumura et al., 2022), while SCRIMP (Wang et al., 2023) uses Transformer-based communication for local tie-breaking. But these reactive methods focus strictly on immediate 1-step collisions. Lacking global topological awareness, they often succumb to oscillatory failures in deep local minima

(e.g., U-shaped traps or long corridors). Recent analysis (Jain et al., 2026) further attributes such failures to weak pairwise interaction biases, not only to short horizons.

## 2.2 Learning-based Foundation Models

The success of large language models inspired the development of "foundation models" for MAPF. MAPF-GPT (Andreychuk et al., 2025b) and MAPF-GPT-DDG(Andreychuk et al., 2025a) demonstrate that scaling decision transformers on massive datasets (over 1 billion expert trajectories) can yield expert-level performance with zero-shot generalization capabilities. These models effectively learn to imitate the internal logic of centralized solvers like LaCAM entirely end-to-end. But they inherently rely on fully known environments for global heuristics. LENS explicitly decouples this problem: learning gradients via U-Net and delegating collision validity to search, achieving comparable performance with $< 0.2\%$ of MAPF-GPT's training data.

## 2.3 Potential-Field Learning for Navigation

Traditional artificial potential fields(Kim & Khosla, 1992; Daily & Bevly, 2008) guide agents via superimposed attractive and repulsive gradients, but are prone to local minima and offer no collision-free guarantees. Recent work embeds potential field optimization into neural architectures to address this(Rasekhipour et al., 2016; Lee & Chung, 2024). Notably, GPFNN (Lyu et al., 2025) unfolds potential field iterations as a deep feedforward network, guaranteeing monotonic potential flows over a fully known global graph with predefined boundary conditions. LENS takes a fundamentally different approach: rather than solving global Laplacian operators over a known topology, the Neural Proposer infers potential field gradients directly from partial ego-centric observations, enabling real-time guidance in strictly unknown environments.

## 2.4 Hybrid Search-Learning Frameworks

Hybrid approaches seek to combine the completeness of classical search algorithms with the computational speed of learning models(Veerapaneni et al., 2024; Liu et al., 2020a; Yan & Wu, 2024). For instance, LNS2+RL (Wang et al., 2025) integrates Multi-Agent Reinforcement Learning (MARL) into the Large Neighborhood Search (LNS) framework. In this paradigm, the global navigation framework remains search-based, while learning is utilized strictly as a local repair operator to resolve complex neighborhood conflicts. Similarly, Real-Time LaCAM (Liang et al., 2025) adapts centralized search for real-time execution but inherently relies on maintaining a global search tree to ensure completeness.

Recent hybrid approaches have explored deep learning-guided global search (e.g., Neural Neighborhood Search (Yan & Wu, 2024)), learned representations for fixed planners (Captari & van Hoof, 2025), and learned local graph-routing (van Knippenberg et al., 2021). LENS diverges from these conventional architectures by introducing a specific learned-proposal / bounded-repair interface. Rather than relying on global search heuristics or end-to-end discrete policies, we strictly treat learning as a continuous local topo-metric estimator from partial observations. Concurrently, we delegate the discrete combinatorial logic required for collision avoidance to localized search (L-CBS (Sharon et al., 2015)). This enforces strict collision-free guarantees within bounded spatio-temporal scopes without the overhead of global replanning.

## 2.5 Coordination in Unknown Environments versus Active Exploration

Our setting focuses on goal-directed navigation under partial observability, distinct from multi-robot exploration where maximizing map coverage is the primary objective(Tan et al., 2024; Bramblett et al., 2022; Chen et al., 2019). In LENS, map discovery is a passive byproduct of navigation. To isolate the fundamental challenge of structured deadlock resolution from the orthogonal complexities of SLAM and asynchronous map merging(Bailey et al., 2006; Burgard et al., 2000), we assume a synchronized, shared map ($\mathcal{M}_{shared}$). This abstraction, common in collaborative robotic deployments, ensures that evaluated performance disparities stem directly from topological reasoning and conflict resolution rather than map synchronization logic.

# 3 Problem Formulation

We formulate our setting as the Centralized Collaborative Partially Observable Multi-Agent Path Finding (PO-MAPF) problem. While standard Multi-Agent Reinforcement Learning (MARL)(Skrynnik et al., 2024a; Wong et al., 2021; Samvelyan et al., 2019) paradigms typically focus on fully decentralized policies with learned peer-to-peer communication, our setting abstracts away decentralized map-merging via a synchronized edge server. This allows us to mathematically model the system as a Centralized Collaborative Multi-Agent Partially Observable Markov Decision Process (MPOMDP), formally defined by the tuple $\langle \mathcal{I}, \mathcal{S}, \mathcal{A}, \mathcal{T}, \Omega, \mathcal{O}, \mathcal{R} \rangle$.

**System & State Space** $(\mathcal{I}, \mathcal{S}, \mathcal{A}, \mathcal{T})$**:** The environment is a hidden grid graph $G = (V, E)$. The system consists of $N$ agents $\mathcal{I} = \{n_1, ..., n_N\}$. The joint state $S_t \in \mathcal{S}$ comprises the positions of all agents, where each agent $n_i$ has a current state $s_t^i \in V$ and a global goal $g^i \in V$. At each timestep $t$, an agent takes an action $u_t^i \in \mathcal{A} = \{$Up, Down, Left, Right, Wait$\}$. The transition function $\mathcal{T} : \mathcal{S} \times \mathcal{A}^N \to \mathcal{S}$ is deterministic but strictly constrained by MAPF collision rules: a valid joint transition must avoid Vertex Conflicts (agents $i \neq j$ occupy the same state $s_{t+1}^i = s_{t+1}^j$) and Edge Conflicts (agents traverse the same edge in opposite directions, $s_t^i = s_{t+1}^j$ and $s_{t+1}^i = s_t^j$).

**Observation Space & Collaborative Mapping** $(\Omega, \mathcal{O})$**:** Unlike classical MAPF, the static environment vertices $V$ are initially unknown. The observation space $\Omega$ and observation function $\mathcal{O}$ dictate that each agent $n_i$ receives a local observation $o_t^i \sim \mathcal{O}(S_t) \in \Omega$, which is strictly a limited FOV $L \times L$ ($11 \times 11$) egocentric occupancy grid capturing visible static obstacles and dynamic peers. Agents maintain a synchronized connection to an edge server, reporting $o_t^i$ to incrementally update a shared spatial map $\mathcal{M}_{shared}$ consisting of known free space, obstacles, and unknown frontiers.

**Objective** $(\mathcal{R})$**:** The global objective is to find a joint policy $\pi$ that navigates all agents to their respective $g^i$ while minimizing the expected Sum-of-Costs (SoC), $\sum_{i \in \mathcal{I}} \text{cost}(\tau_i)$ (which corresponds to maximizing the cumulative reward $\mathcal{R}$), subject to the strict safety constraints of avoiding vertex and edge conflicts.

**The Navigation Decomposition:** The navigation framework $\pi$ is a hierarchical decomposition of three sub-functions, designed to bridge local perception and centralized coordination: **1. Global Guidance** $(f_{guide})$**:** Since local views lack topological context (e.g., U-shaped traps), a high-level function is needed to estimate the global gradient: $\mathbf{v}_{guide} = f_{guide}(\mathcal{M}_{shared}, g^i)$. **2. Local Proposal** $(f_{prop})$**:** Agents must react to immediate dynamic obstacles. A neural proposer approximates local value function (cost-to-go potential field) to generate a motion intent: $\tilde{u}_t^i = f_{prop}(o_t^i, \mathbf{v}_{guide})$. **3. Conflict Resolution** $(f_{defense})$**:** Since myopic neural proposals are prone to deadlocks, the central server monitors proposed trajectories for anticipated conflicts. If structural deadlocks or collisions $\mathcal{I}_{conf}$ are detected, the server overrides local proposals with a coordinated centralized search: $\mathbf{u}_t = f_{defense}(\mathcal{M}_{shared}, \tilde{\mathbf{u}}_t)$, where $\tilde{\mathbf{u}}_t$ represents the joint neural proposals.

Our objective is to learn and engineer these three components to minimize the makespan, enforcing local collision avoidance within the Defense Cascade's resolution scope, without access to a pre-computed global oracle. Note that LENS targets goal-directed navigation, not map exploration. Map discovery is a passive byproduct, not an objective. We abstract away SLAM and decentralized map merging via a synchronized shared map ($\mathcal{M}_{shared}$), isolating topology-aware conflict resolution as the core challenge.

## 4  Method: The LENS Framework

As shown in Figure 1, the LENS inference pipeline instantiates the MPOMDP components defined in Section 3 through three coupled modules. First, the *Adaptive Guidance Mechanism* incrementally recovers the hidden graph $G$ to extract high-level subgoals ($\mathbf{v}_{guide}$). Second, the *Neural Proposer* acts on local observations $o_t^i \in \Omega$ to regress a dense potential field $\hat{\Phi}$ (an approximate local value function) for generating multi-step motion intents. Finally, the *Defense Cascade* serves as the strict enforcer of the transition constraints $\mathcal{T}$; by executing bounded search over anticipated conflicts, it outputs the final collision-free joint actions $\mathbf{u}_t$.

### 4.1  The Neural Proposer: Heuristic Approximation via U-Net

The core decision-making module is the *Neural Proposer*. Unlike standard policies that classify discrete actions directly, which often discard spatial relativity, we formulate the learning task as **Heuristic Ap-**

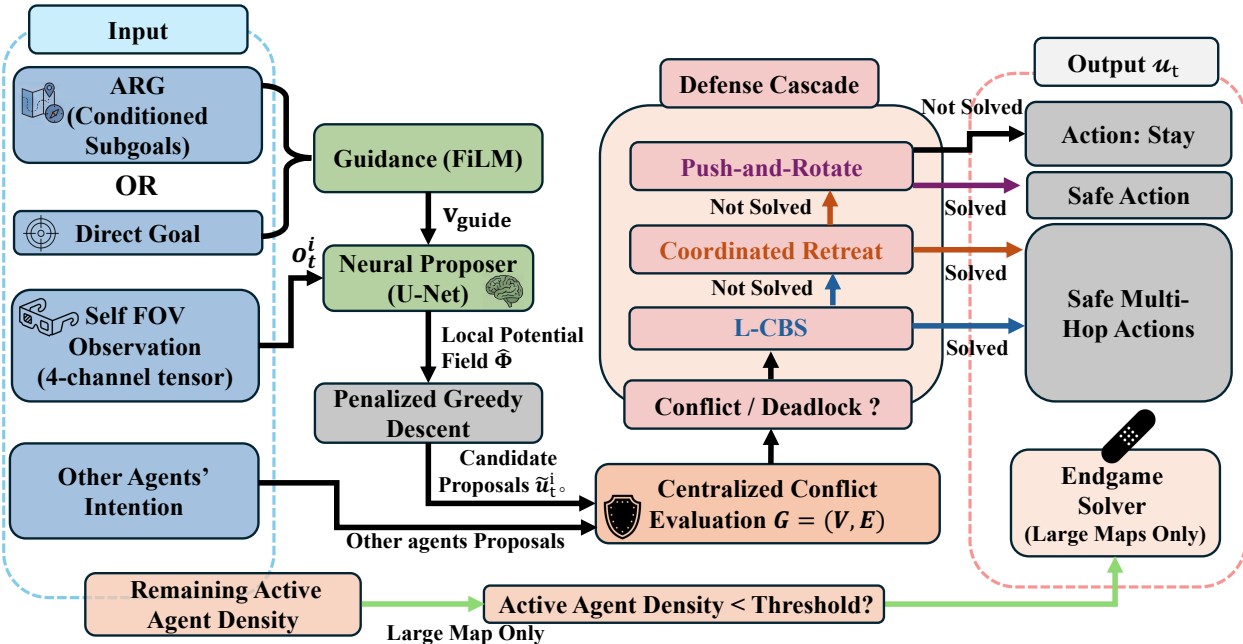

Figure 1: **The LENS Execution Flowchart.** The framework clearly maps distinct Inputs (local observations, ARG subgoals, peer intents) to safe Output actions. The Neural Proposer fuses the guidance vector with local observations to regress a dense potential field. If centralized evaluation anticipates conflicts within these proposals, the Defense Cascade is triggered to guarantee collision-free outputs. Arrow refers to data flow. The blue blocks represent the raw state inputs $\Omega$ and guidance, while the grey blocks represent the final collision-free action outputs $u_t$. The entire pipeline depicted operates during the online inference phase.

**proximation**. The network learns to approximate the local crop of the ground-truth global potential field $\Phi^*$ within the agent's field of view.

We employ a U-Net architecture for three geometrically motivated reasons. First, Navigation is an inherently geometric problem. To preserve spatial topology, U-Net's skip connections preserve high-frequency spatial details (e.g., obstacle boundaries) that are essential for collision avoidance but often lost in the bottleneck layers of standard CNNs. Second, the output is a *Cost-to-Go Potential Field* $\hat{\Phi} \in \mathbb{R}^{L \times L}$. This scalar field provides a smooth gradient of descent, which is an approximate local value function of an optimal global policy, allowing agents to flow around obstacles by following the path of least resistance. Third, to effectively integrate heterogeneous inputs (static walls vs. dynamic agents), we enhance the U-Net encoder with **Squeeze-and-Excitation (SE) Blocks** (Hu et al., 2018), enabling adaptive channel-wise feature recalibration to fuse context via attention.

**Goal Conditioning:** The network input $o_t^i$ comprises a 4-channel local spatial tensor (Obstacles, Dynamic Agents, Self-Position, Projected Local Target) and a non-spatial global guidance vector $\mathbf{v}_{guide}$. To effectively integrate the micro-scale spatial perception with the macro-scale topological intent, we utilize Feature-wise Linear Modulation (FiLM) (Brockschmidt, 2020) at the bottleneck. The non-spatial vector modulates the feature maps via affine transformations, effectively conditioning the predicted potential field on the global navigation intent. The detailed architecture is provided in Appendix F.1.

**Action Extraction: Discrete Potential Descent:** Given the predicted local potential field $\hat{\Phi}$, the agent performs a masked greedy descent to identify a short-term waypoint (local subgoal) within the prediction horizon. Geometric obstacles act as hard constraints. For any valid transition $p' \in \mathcal{N}(s_t^i) \setminus \mathcal{O}_{local}$, where $\mathcal{N}(s_t^i)$ denotes the 4-connected neighbors of $s_t^i$ and $\mathcal{O}_{local}$ is the set of occupied cells in the FOV, the agent minimizes the adjusted potential $J(p')$: $J(p') = \hat{\Phi}(p') + \psi(p', H_{local})$ s.t. $p' \notin \mathcal{O}_{local}$ where $\mathcal{O}_{local}$

represents the set of occupied cells in the current local field-of-view. The history-dependent term $\psi$ acts as an anti-patterning regularizer to suppress oscillatory behavior:

$$\psi(p') = \begin{cases} \lambda_{visit} & \text{if } p' \in H_{local} \text{ (recently visited)} \\ \lambda_{stay} & \text{if } p' = s_t^i \text{ and } s_t^i \neq g^i \\ -\lambda_{mom} & \text{if } (p' - s_t^i) = \mathbf{d}_{t-1}^i \text{ (momentum)} \\ 0 & \text{otherwise} \end{cases} \tag{1}$$

Weights tuned via grid search on a hold-out validation set ($\lambda_{visit} = 0.5, \lambda_{stay} = 0.6, \lambda_{mom} = 0.2$) ensure smooth trajectories while discouraging local livelocks. Moving direction $\mathbf{d}_{t-1}^i = s_t^i - s_{t-1}^i$.

**Data Generation and Training Strategy:** We train the U-Net in a supervised manner, treating the problem as heuristic reconstruction: learning to predict the "ideal" navigation potential from local views, rather than cloning discrete actions from an expert policy. Training hyperparameters are comprehensively detailed in Appendix G.

Step 1. **Trajectory Simulation and Targeted Sampling:** We simulate MAPF episodes using **LaCAM** (Okumura, 2023) solely as a generative engine to produce diverse, valid agent configurations representative of different scenarios, not as a policy to imitate. We apply event-based filtering to focus on two critical navigation phases: *initialization* (starting configuration) and *terminal approach* (within 5 steps of the goal), where the underlying topology is most distinct.

Step 2. **Ground Truth Potential Field Computation:** For each sampled state, we compute a global ground truth potential field via backward BFS from the agent's goal $g_i$ on the fully known map. Each cell value encodes the true shortest distance to the goal, serving as the target geometric gradient.

Step 3. **Training Sample Assembly:** The input $o_t^i$ is the agent's local FOV paired with a normalized goal vector injected into U-Net features via a **FiLM** layer. The target label $\Phi^*$ is the corresponding FOV-sized patch of the global BFS potential field. A detailed visual breakdown of the raw 4-channel input tensors, the global geometric context, and the corresponding local prediction labels is provided in Appendix G (Figure G.2).

**Composite Loss Function:** To ensure the predicted potential field $\hat{\Phi}$ captures both accurate distance values and the correct local gradient shape (which determines motion direction), we train the model using a composite loss function $\mathcal{L}_{total}$:

$$\mathcal{L}_{total} = \mathcal{L}_{MSE} + \lambda_{grad}\mathcal{L}_{grad} \tag{2}$$

The primary loss, $\mathcal{L}_{MSE}$, measures the pixel-wise Mean Squared Error between the prediction $\hat{\Phi}$ and the target $\Phi_{target}$ (Hodson, 2022). The auxiliary gradient loss, $\mathcal{L}_{grad}$, penalizes discrepancies in spatial gradients (approximated via Sobel filters):

$$\mathcal{L}_{grad} = \|\nabla_x\hat{\Phi} - \nabla_x\Phi_{target}\|_1 + \|\nabla_y\hat{\Phi} - \nabla_y\Phi_{target}\|_1 \tag{3}$$

This gradient-aware objective (Chen et al., 2018) encourages the model to preserve the steepness and directionality of the potential field, which is critical for the subsequent greedy path extraction.

## 4.2 The Defense Cascade

While the Neural Proposer handles most steps efficiently, unguided local intent inevitably leads to multi-agent conflicts. LENS operates on a receding horizon control (RHC) scheme: proposing paths for $k$ steps (e.g., $k = \min(d_{\text{subgoal}}, 10)$) but executing only the first $n$ steps (e.g., $n = 3$ or $n = 6$). Before execution, the server constructs a Spatio-Temporal Conflict Graph from the joint proposals and extracts disjoint connected components. We introduce the **Defense Cascade**, applied independently to each disjoint conflict group. This isolation strictly prevents interference between overlapping resolution windows.

**Deadlock Entry Zone Detector (DEZD):** The DEZD module continuously monitors the agent's short-term positional history queue $H_{t-k:t}^i$ of length $k$. Rather than computing computationally expensive spatial

bounding boxes or sequence overlaps, we employ a strict and lightweight entropy-based heuristic: a deadlock is flagged if the number of unique spatial coordinates visited within the history window falls below a critical threshold $\tau_{unique}$. The intervention is triggered when $|\text{Set}(H_{t-k:t}^i)| \leq \tau_{unique}$, effectively capturing both extreme positional stagnation and tight cyclic oscillations (e.g., $k = 10, \tau_{unique} = 3$). These specific threshold values were empirically determined via a grid search ($k \in [5, 10], \tau_{unique} \in [2, 6]$) on a hold-out validation set to optimally balance false positives (triggering computationally expensive search for temporary congestion) against false negatives (wasting execution steps idling in hard deadlocks).

**The Hierarchy of Resolution:** The cascade consists of strategies ordered by computational cost and "aggressiveness" (see Figure 1). Implementation details for each level are provided in Appendix H.

**1. L-CBS:** The foundational defense layer. To ensure strict safety, the server first constructs a Spatio-Temporal Conflict Graph $\mathcal{G}_{conf}$ from the neural proposals, where edges denote anticipated vertex or edge collisions within a $k$-step horizon. We partition the agents into independent sub-problems by extracting the Connected Components of $\mathcal{G}_{conf}$, forming strictly disjoint conflict groups. For each group, L-CBS operates utilizing Space-Time $A^*$ as the low-level planner, treating anticipated interactions as hard spatio-temporal constraints. Following the properties of standard CBS (Sharon et al., 2015; Boyarski et al., 2015), the search is complete and optimal within its temporal bounds. By enforcing a receding horizon—planning for $k$ steps (with a terminal "Wait" condition) but executing only the first $n$ steps ($n < k$)—we establish a temporal safety buffer. This guarantees that the executed $n$-step multi-agent trajectories are strictly free of vertex or edge collisions, effectively bounding the local execution safety.

**2. Coordinated Retreat via Distance-based Symmetry Breaking.** When L-CBS times out, agents are partitioned into two roles by heuristic distance: *Yielders* (furthest from their goals) run BFS to find the nearest free retreat cell, while *Movers* re-run L-CBS with Yielders' planned trajectories masked as dynamic constraints.

**3. Push-and-Rotate:** For small, tightly interlocked groups (e.g., a $2 \times 2$ cycle), this heuristic executes a synchronized rotation of agents into adjacent empty cells.

**4. Endgame Solver (Large-Scale Exclusive):** In massive environments like City maps, a few remaining agents (stragglers) may wander indefinitely after the majority reach their goals. When active agent density drops below a critical threshold (e.g., 15%), LENS triggers a globally coherent CBS exclusively for these stragglers to guarantee episode termination.

By prioritizing L-CBS, we maintain solution quality; by falling back to heuristic maneuvers, we guarantee robustness against catastrophic deadlocks. Empirical profiling (detailed in Appendix D) validates this hierarchical design: standard L-CBS successfully resolves over 98.9% of conflict groups, confirming that the heuristic fallbacks act strictly as rare safety nets rather than the primary driver of resolution.

### 4.3 Collaborative Mapping and Adaptive Guidance

To overcome the myopia of local observations, LENS operates in a collaborative mapping and planning paradigm. Agents maintain a synchronized connection to an edge server via perfect broadcast communication, incrementally constructing a unified Persistent Known Map $\mathcal{M}_{shared}$ (Lajoie et al., 2022). While neural proposals can be generated efficiently using local views, the server acts as the central orchestrator for the Defense Cascade, executing L-CBS directly on the synchronized $\mathcal{M}_{shared}$ to untangle overlapping agent intents. This setup abstracts away communication latency and fully decentralized SLAM overheads, allowing us to strictly isolate the core algorithmic challenge: resolving structural deadlocks via online topological discovery rather than relying on pre-computed global oracles.

Based on $\mathcal{M}_{shared}$, the system maintains a global topological abstraction, the ARG $G_{ARG} = (V_{reg}, E_{reg})$, as a singleton instance shared across all agents. $V_{reg}$ consists of fixed-size grid regions (e.g., $16 \times 16$ cells). The connectivity set $E_{reg}$ is updated incrementally. An edge $e_{ij}$ between regions $r_i$ and $r_j$ is considered *traversable* only if the boundary cells in $\mathcal{M}_{shared}$ are effectively free. For cells within $\mathcal{M}_{shared}$ marked as *Unknown*, the ARG adopts an optimistic assumption, treating them as traversable. While this encourages active exploration of frontier regions, it inherently risks guiding agents into large topological traps (e.g., deep dead-ends) as map discovery unfolds. LENS mitigates this structural vulnerability through a tight feedback

loop with the DEZD. In Ablation Study (Table 4), disabling this loop in unknown environments causes the Success Rate to drop by 6% and the Sum of Costs to surge by nearly 1000, since agents stagnate in unpredicted dead-ends.

**Guidance Generation:** As agents update $\mathcal{M}_{shared}$ with new observations $o_t^i$, the ARG re-evaluates connectivity. A high-level search on the ARG extracts a valid macroscopic subgoal. This subgoal (or its short guiding trajectory) is then directly projected into the agent's ego-centric FOV to form the fourth spatial input channel. This ensures that distant agents immediately benefit from structural discoveries (e.g., newly blocked corridors) made by their peers (Zhang et al., 2024).

## 5 Experimental Evaluation

Our evaluation strategy is designed to decouple the two core capabilities of the LENS framework: (1) the *architectural efficiency* of the potential-guided search when topological information is accurate, and (2) the *inference robustness* of the neural proposer when operating in unknown environments. To this end, we conduct experiments under two rigorously defined protocols: **Setting A (Architecture Validation)** and **Setting B (Unknown Environment Exploration)**.

### 5.1 Experimental Setup

**Benchmark Environments:** We utilize the POGEMA benchmark (Skrynnik et al., 2025), covering four distinct topologies: (1) **Random:** Unstructured clutter with 30% obstacle density. (2) **Mazes:** Procedurally generated corridors and dead ends requiring long-horizon reasoning. (3) **Warehouse:** Structured logistics layouts testing extreme agent density. (4) **Cities:** Large-scale urban maps from MovingAI, testing scalability. Complete details regarding map configurations, hardware specifications, and all system hyperparameters (e.g., DEZD trigger thresholds) are provided in Appendices G, F, and I.

**Baselines & Adaptations:** We evaluate LENS against state-of-the-art methods across different paradigms: **LaCAM** (Okumura, 2023) (centralized oracle), **SCRIMP** (Wang et al., 2023) (communication MARL), **LNS2-RL** (Li et al., 2022a) (hybrid search-learning), **CS-PIBT** (Veerapaneni et al., 2025) (reactive heuristic), and **MAPF-GPT** (Andreychuk et al., 2025b) (foundation model). To ensure a fair comparison in strictly unknown environments (Setting B), applicable baseline algorithms were adapted to integrate with an online incremental map discovery framework. Comprehensive descriptions, training dataset setups, and adaptation mechanics for all baselines are deferred to Appendix E.

**Evaluation Metrics:** We report four standard metrics to evaluate both system-level and agent-level performance: (1) *Success Rate (SR)*, the proportion of evaluated maps where all agents successfully reach their destinations within the step limit without any collisions; (2) *Individual Success Rate (ISR)*, the average percentage of individual agents that successfully reach their respective goals within a map, which provides a granular measure of partial progress in highly congested scenarios; (3) *Makespan*, the maximum number of timesteps required for the last agent to reach its goal; and (4) *Sum of Costs (SoC)*, the cumulative number of timesteps taken by all agents. Crucially, as noted in the figure and table captions, Makespan and SoC are computed strictly over *fully successful instances* (where SR applies) to avoid survivorship bias.

### 5.2 Setting A: Architecture Validation (Global Knowledge)

**Objective:** To evaluate the quality of the Neural Proposer's learned heuristic approximation against oracle-guided baselines. We note that Setting A is asymmetric in information (learned inference vs. oracle heuristics) and serves primarily as an *architecture validation* to prove the U-Net extracts useful topology, rather than a perfectly fair system-level comparison. In this setting, all methods have access to the static global map. However, this setup is deliberately asymmetric: while the baseline methods receive pre-computed ground-truth cost-to-go maps (i.e., oracle heuristic guidance), LENS's U-Net performs learned inference from perception. This asymmetric setup places LENS at a relative disadvantage, as baselines are provided with ground-truth heuristics while LENS relies entirely on its learned potential field approximation. We emphasize that the U-Net receives no ground-truth potential field in Setting A; all guidance is derived from

Table 1: **Small-scale Environments.** ISR and Makespan comparisons on Maze and Random maps.

| | Maze Maps | | | | | | | | | | | | Random Maps | | | | | | | | | | | |
|---|---|---|---|---|---|---|---|---|---|---|---|---|---|---|---|---|---|---|---|---|---|---|---|---|
| | ISR | | | | | | Makespan | | | | | | ISR | | | | | | Makespan | | | | | |
| Agents | 8 | 16 | 24 | 32 | 48 | 64 | 8 | 16 | 24 | 32 | 48 | 64 | 8 | 16 | 24 | 32 | 48 | 64 | 8 | 16 | 24 | 32 | 48 | 64 |
| LaCAM | 1.00 | 1.00 | 1.00 | 1.00 | 1.00 | 1.00 | 29.1 | 33.3 | 37.8 | 40.0 | 50.0 | 61.9 | 1.00 | 1.00 | 1.00 | 1.00 | 1.00 | 1.00 | 24.0 | 27.2 | 29.0 | 30.7 | 33.3 | 36.0 |
| SCRIMP | _0.98_ | 0.88 | 0.82 | 0.68 | 0.50 | 0.37 | 42.7 | 64.5 | 109.3 | 99.2 | 114.2 | - | 0.97 | 0.90 | 0.80 | 0.71 | 0.67 | 0.64 | 43.4 | 74.2 | 104.1 | 132.7 | 121.7 | 123.2 |
| CS-PIBT | **1.00** | **1.00** | _0.99_ | _0.99_ | **0.99** | **0.91** | 34.0 | _35.0_ | **38.0** | **52.0** | 101.0 | _101.6_ | **1.00** | **1.00** | _0.99_ | _0.99_ | _0.99_ | **0.99** | 26.7 | 31.5 | _36.1_ | 42.4 | 57.9 | 89.9 |
| LNS2-RL | 0.98 | _0.99_ | 0.98 | 0.98 | 0.94 | 0.83 | 39.7 | 51.6 | 62.0 | 66.4 | **75.0** | **72.3** | _0.99_ | _0.99_ | **1.00** | _0.99_ | 0.98 | 0.95 | 31.7 | 36.8 | 42.3 | 45.6 | _53.7_ | **58.9** |
| GPT-2M | **1.00** | _0.99_ | 0.98 | 0.94 | 0.81 | 0.65 | 37.8 | 60.0 | 85.2 | 96.1 | 109.5 | 126.4 | **1.00** | **1.00** | **1.00** | **1.00** | _0.99_ | 0.92 | _26.2_ | 33.1 | 37.8 | 46.3 | 63.8 | 92.7 |
| GPT-6M | **1.00** | **1.00** | **1.00** | 0.99 | 0.94 | 0.83 | 33.6 | 44.1 | 54.7 | 72.6 | 97.7 | 128.1 | **1.00** | **1.00** | **1.00** | **1.00** | **1.00** | 0.96 | **26.1** | _31.4_ | 36.6 | _40.7_ | 58.2 | 82.1 |
| GPT-85M | **1.00** | **1.00** | **1.00** | **1.00** | _0.96_ | _0.88_ | _32.2_ | 39.6 | 47.6 | 58.5 | _86.3_ | 121.7 | **1.00** | **1.00** | **1.00** | **1.00** | **1.00** | _0.98_ | 26.4 | 33.1 | **35.2** | 39.8 | **52.5** | _72.2_ |
| LENS | **1.00** | **1.00** | _0.99_ | 0.99 | 0.95 | 0.81 | **31.0** | **33.4** | _43.9_ | _57.0_ | 102.6 | 169.2 | **1.00** | **1.00** | _0.99_ | 0.98 | 0.98 | 0.96 | 27.6 | **29.4** | 36.2 | 45.1 | 66.9 | 85.6 |

learned inference. This distinguishes Setting A from a trivial oracle upper bound and validates the quality of the Neural Proposer as a heuristic approximator.

**Results Analysis:** Figure 2 presents a comprehensive comparison of Success Rate (SR), Runtime, and Sum of Costs (SoC). Detailed numerical results are provided in Tables 1 and 2 (Best in **Bold**, Second Best in _Underlined italic_.) Makespan/ISR values are computed only over fully successful instances; in low-SR regimes, they reflect survivorship bias.

**Robustness in Structured Environments:** As detailed in Table 1 and Figure 2a, LENS substantially outperforms all decentralized baselines in structured topologies (e.g., yielding more than double the Success Rate of CS-PIBT and LNS2-RL in Maze-64). It maintains competitive robustness relative to the centralized oracle LaCAM. This confirms that while reactive heuristics struggle with structured deadlocks despite having oracle global guidance, the bounded search within the Defense Cascade efficiently resolves these deep topological traps.

**Efficiency at Scale:** Regarding **Runtime** (Figure 2, Row 2), LENS exhibits superior scalability even while performing active neural inference rather than relying on pre-computed heuristics. In **Random** maps, **LENS** is approximately 2× faster than LNS2-RL and 5× faster than MAPF-GPT-85M. In the extreme density of the **Warehouse** (192 agents), LENS solves instances in 64s, whereas MAPF-GPT requires 242s. While LENS incurs higher total runtime in massive topologies like **Cities** (256 agents) compared to purely reactive heuristics, a granular computational profile (Appendix D) reveals this is an explicit engineering trade-off: the U-Net inference and L-CBS solver combined consume around 5% of the latency. The overhead stems primarily from the continuous synchronization of the global dynamic map across 256 agents, an aspect independent of the core decision-making efficiency. These profiles validate that the core decision stack itself (U-Net + L-CBS) is highly lightweight, while the end-to-end system latency in massive regimes is heavily dominated by the engineering overhead of centralized state synchronization.

**Solution Quality:** The **SoC** results (Figure 2, Row 3) indicate that LENS generates paths of comparable quality to LaCAM, despite LENS receiving no oracle guidance. Notably, in Random maps, **LENS** achieves a lower SoC than other learning-based methods equipped with ground-truth BFS, suggesting that the learned potential field effectively approximates the true shortest path.

### 5.3 Setting B: Robustness in Unknown Environments (online collaborative mapping)

**Objective:** This setting mimics realistic deployments where pre-computation is impossible. Unlike Setting A, the global map is initially hidden. To rigorously evaluate our collaborative framework, we enforce an incremental map discovery protocol: agents navigate strictly using local FOVs without any prior map knowledge, while all valid observations are instantly merged into a shared dynamic map $\mathcal{M}_{shared}$ to progressively reveal the topology. All methods in Setting B can access the synchronized shared map ($\mathcal{M}_{shared}$), updated identically via agents' collective observations.

**Comprehensive Performance Analysis:** Since all compared methods share identical map information in Setting B, the observed performance differences reflect purely algorithmic advantages in topological reason-

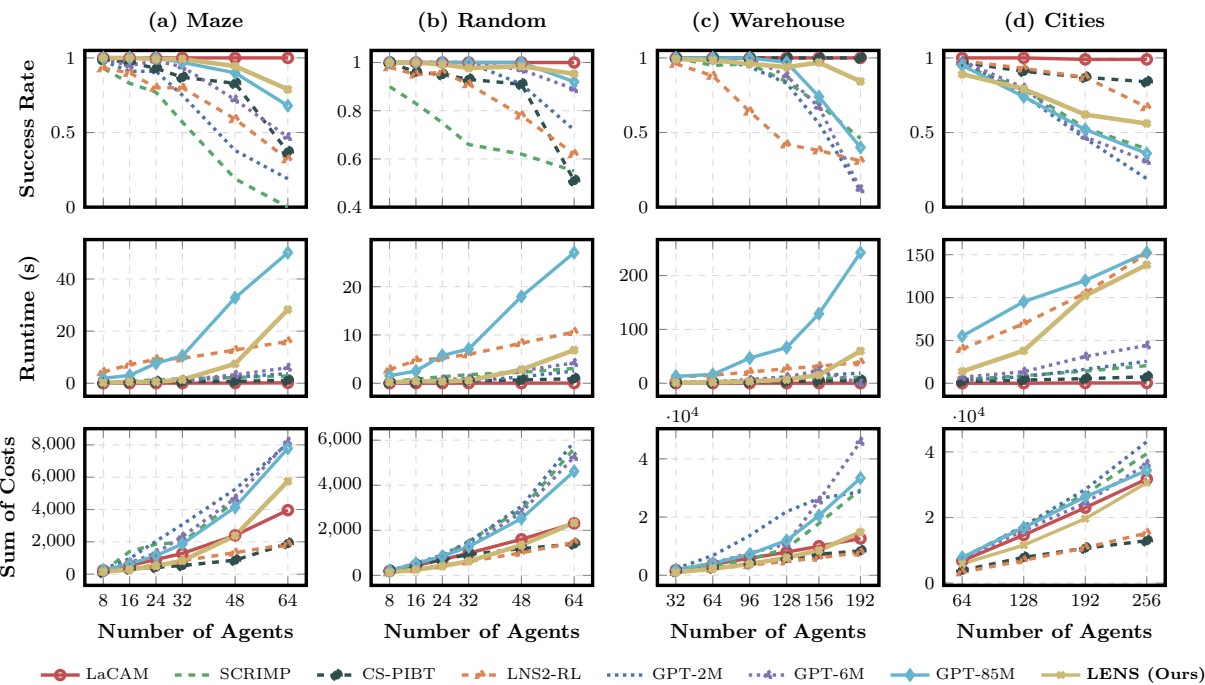

Figure 2: **Comprehensive Performance (Setting A).Top Row:** SR (Higher is better). **Middle Row:** Runtime (Lower is better). **Bottom Row:** SoCs (Lower is better). **Legend:** LENS (Gold) demonstrates superior robustness in high-complexity maps (Maze/Warehouse) compared to reactive and learning-based baselines. Runtime and SoCs are averaged over successful runs. Points are omitted when an algorithm produces no successful runs in that setting; thus, the SoCs is undefined rather than missing.

Table 2: **Large-scale Environments.** ISR and Makespan comparisons on Warehouse and City maps.

| | Warehouse | | | | | | | | | | | | Cities | | | | | | | |
|---|---|---|---|---|---|---|---|---|---|---|---|---|---|---|---|---|---|---|---|---|
| | ISR | | | | | | Makespan | | | | | | ISR | | | | Makespan | | | |
| Agents | 32 | 64 | 96 | 128 | 156 | 192 | 32 | 64 | 96 | 128 | 156 | 192 | 64 | 128 | 192 | 256 | 64 | 128 | 192 | 256 |
| LaCAM | 1.00 | 1.00 | 1.00 | 1.00 | 1.00 | 1.00 | 55.2 | 58.9 | 60.7 | 62.0 | 63.9 | 65.7 | 1.00 | 1.00 | 0.99 | 0.99 | 105.7 | 114.4 | 119.7 | 123.9 |
| SCRIMP | 0.98 | 0.96 | _0.99_ | 0.93 | 0.83 | 0.72 | 109.1 | 156.7 | 186.5 | 172.4 | 181.0 | 192.0 | **1.00** | 0.95 | 0.85 | 0.78 | 174.5 | 152.2 | 168.2 | 182.6 |
| CS-PIBT | **1.00** | **1.00** | **1.00** | **1.00** | **1.00** | **1.00** | 62.0 | 69.5 | 82.0 | 104.0 | 138.0 | _164.0_ | **1.00** | **1.00** | 0.99 | 0.99 | 118.2 | **123.6** | 129.4 | _134.6_ |
| LNS2-RL | **1.00** | **1.00** | _0.99_ | 0.98 | _0.98_ | _0.98_ | 66.7 | 68.2 | **72.5** | **73.2** | **76.6** | **79.0** | **1.00** | **1.00** | 0.99 | _0.98_ | **115.6** | _124.1_ | _128.6_ | **130.0** |
| GPT-2M | **1.00** | **1.00** | 0.97 | 0.80 | 0.68 | 0.42 | 70.3 | 105.7 | 142.0 | 171.0 | 214.5 | 236.2 | _0.99_ | _0.96_ | 0.89 | 0.80 | _116.5_ | 135.8 | 148.7 | 168.3 |
| GPT-6M | **1.00** | **1.00** | **1.00** | 0.96 | 0.95 | 0.79 | **57.2** | _64.9_ | 79.1 | _83.8_ | 166.8 | 239.0 | **1.00** | 0.95 | 0.89 | 0.83 | 118.0 | 125.0 | **127.3** | 141.9 |
| GPT-85M | **1.00** | **1.00** | **1.00** | _0.99_ | 0.87 | 0.78 | _60.2_ | 64.6 | _75.1_ | 93.4 | _133.1_ | 174.0 | _0.99_ | 0.95 | 0.87 | 0.88 | 122.0 | 132.1 | 136.9 | 143.1 |
| LENS | _0.99_ | _0.98_ | 0.96 | 0.94 | 0.97 | 0.87 | 63.3 | 89.3 | 116.6 | 154.7 | 182.1 | 233.0 | 0.97 | 0.95 | _0.93_ | 0.89 | 243.1 | 224.1 | 253.2 | 293.1 |

ing and deadlock resolution. Figure 3 visualizes performance in strictly unknown environments. Reactive schemes like CS-PIBT fundamentally lack the horizon to foresee structural deadlocks (e.g., entering a corridor that is blocked at the other end). In Setting B, we compare only against baselines portable to unknown environments (SCRIMP, adapted LNS2-RL, adapted CS-PIBT), as other hybrid solvers require fully known maps.

**The Cost of Myopia:** Without pre-computed flow guidance, **CS-PIBT** (Blue/Dashed) suffers significant performance degradation in structured environments. (1) **Maze (Figure 3a):** Success rate drops to **6.3%** at 64 agents (down from 37.2% in Setting A). Agents oscillate in dead ends. (2) **Warehouse (Figure 3c):** The performance degradation is absolute. At 192 agents, SR drops from 100% (Known) to **0%** (Unknown). Local conflict resolution cannot handle massive congestion in narrow aisles without a global plan. This

collapse reflects both 1-step myopia and limited pairwise attention bias under dense coupling (Jain et al., 2026); LENS mitigates both by coupling a multi-step gradient with L-CBS as a structural bypass.

**The Scalability Wall of Search: LNS2-RL** performs reasonably in Random maps but hits a scalability wall in complex topologies. In **Cities** (Figure 3d), LNS2-RL performance drops to 59.8% at 256 agents with excessive runtime costs ($> 180s$), caused by the need to constantly re-plan as new obstacles are discovered.

**LENS Stability:** LENS demonstrates remarkable stability. Its performance curves in Setting B show minimal degradation compared to Setting A, with success rate gaps consistently smaller than those of all baselines across almost all tested topologies. This empirically proves that LENS does not rely on map memorization but on generalized inference of topology from local views. However, we note that the observed robustness (i.e., the minimal degradation from Setting A to B) is the joint product of the generalized local-topology inference, the synchronized shared-map scaffold, the ARG prior, and the explicit repair stack. While our ablations(Table 4) confirm each component's necessity, the exact contribution of the U-Net's inherent robustness versus the system's structural safeguards remains an entangled property of the hybrid method.

Table 3: **Gap Analysis (Setting B vs. Setting A).** The "Gap" represents $Value_{SettingB} - Value_{SettingA}$. A negative Gap in SR indicates performance loss due to unknown environments.

| Map | Count | CS-PIBT | | LNS2-RL | | LENS (Ours) | |
|---|---|---|---|---|---|---|---|
| | | SR (Gap) | Time /s (Gap) | SR (Gap) | Time /s (Gap) | SR (Gap) | Time /s (Gap) |
| **Maze** | 8 | 79.5% (−19.5) | *1.46 (+1.2)* | *92.9% (+0.0)* | 5.60 (+1.3) | **100% (+0.0)** | **0.59 (+0.4)** |
| | 32 | 28.1% (−58.9) | *6.13 (+5.5)* | *73.4% (−6.6)* | 11.61 (+2.1) | **98.0% (−1.2)** | **4.53 (+3.2)** |
| | 64 | 6.3% (−30.7) | **13.82 (+12.7)** | *38.3% (+6.3)* | *18.11 (+2.1)* | **79.0% (+0.1)** | 35.81 (+7.6) |
| **Random** | 8 | 95.3% (−4.7) | *0.93 (+0.8)* | *96.1% (−1.9)* | 3.98 (+1.1) | **100% (+0.0)** | **0.48 (+0.3)** |
| | 32 | 61.7% (−31.3) | *4.39 (+3.9)* | *93.0% (+2.0)* | 7.76 (+1.8) | **96.0% (−1.7)** | **0.71 (+0.1)** |
| | 64 | 11.8% (−39.2) | 13.96 (+13.0) | *66.4% (+4.4)* | *12.88 (+2.3)* | **91.0% (−4.3)** | **4.76 (−2.1)** |
| **Warehouse** | 32 | 95.3% (−4.7) | 21.47 (+20.6) | *96.1% (+0.1)* | *14.79 (+3.4)* | **100% (+0.8)** | **3.84 (+3.1)** |
| | 128 | 10.9% (−89.1) | *27.52 (+24.4)* | *42.2% (+0.2)* | 34.36 (+7.4) | **97.0% (+3.2)** | **24.39 (+17.7)** |
| | 192 | 0.0% (−100.0) | **39.27 (+34.4)** | *26.6% (−4.4)* | *49.42 (+10.8)* | **64.0% (−20.3)** | 63.93 (+4.0) |
| **City** | 64 | 62.2% (−34.8) | 168.28 (+166.3) | **97.6% (−0.4)** | *61.33 (+21.3)* | *83.0% (−6.1)* | **30.71 (+17.1)** |
| | 128 | 56.7% (−34.3) | 113.12 (+109.4) | **93.7% (+0.7)** | *93.87 (+24.3)* | *77.3% (−1.8)* | **81.43 (+43.4)** |
| | 256 | 28.3% (−55.7) | **139.09 (+131.5)** | **59.8% (−7.2)** | 186.01 (+34.5) | *46.2% (−9.8)* | *184.1 (+27.03)* |

**Gap Analysis: The Cost of Blindness:**

Table 3 quantifies the performance degradation ($\Delta$) when switching from Setting A to Setting B.

**Success Rate Gap:** CS-PIBT suffers massive drops in structured maps ($\Delta_{SR} = −39.2\%$ in Random-64, $−100\%$ in Warehouse-192). LNS2-RL, while more robust, still degrades significantly in dense structured environments ($\Delta_{SR} = −4.4\%$ in Random-64, $−20.3\%$ in Warehouse 192), revealing the scalability limits of search-based repair under unknown topologies.

**Runtime Gap:** While LNS2-RL slows down significantly in Warehouse ($\Delta_{Time} = +34.4s$) due to deadlocks, LENS has a **negligible Gap** ($\Delta \approx 0$), as its architecture is inherently designed for exploration.

## 5.4 Ablation Study

To validate the contribution of each module within the LENS framework, we conducted ablation studies across two regimes: high-density structured deadlocks where the system operates in *Direct Guidance* mode, and large-scale navigation where the system activates *Hierarchical Guidance* (ARG).

**Impact of the Neural Proposer:** A critical finding is the indispensability of the U-Net-based *Neural Proposer*. In the **Maze** environment (Table 4), replacing the U-Net with a standard A* solver ("w/o U-Net") degrades the success rate from **78.7**% to 64.1%. This confirms that the learned potential field captures spatial features of dead-end structures better than generic heuristics.

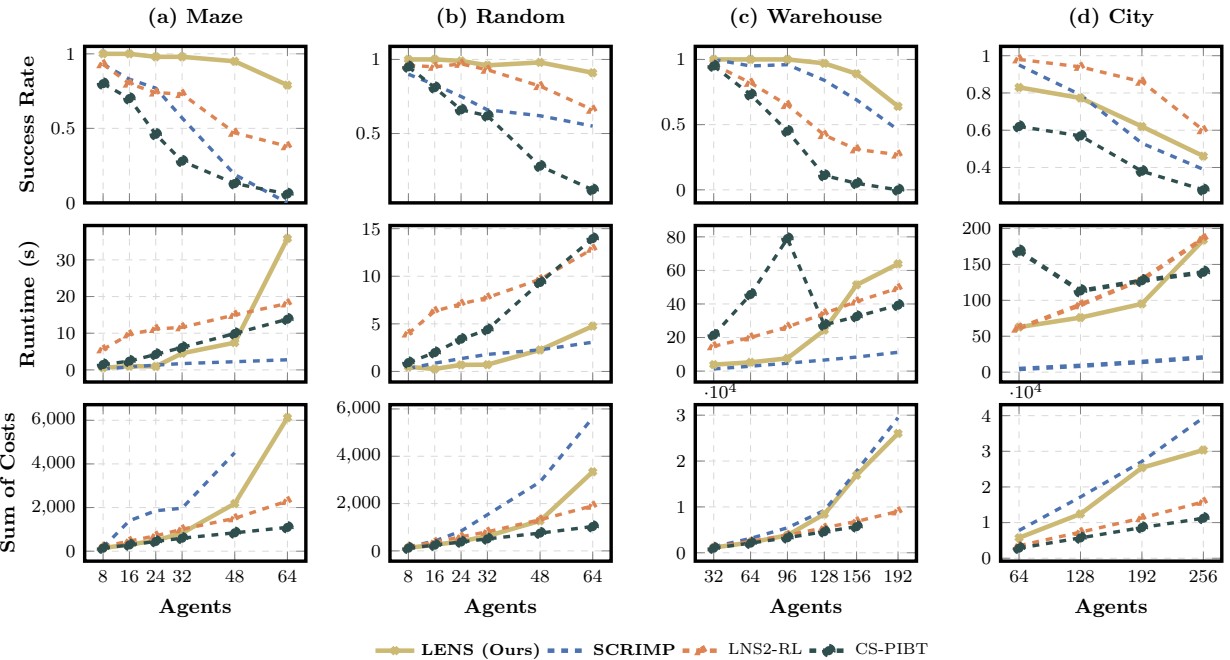

Figure 3: **Performance in Setting B.** Comparing LENS against SCRIMP, LNS2-RL and CS-PIBT when the global map is unknown. **Top Row:** SR. Note the dramatic collapse of CS-PIBT in Maze and Warehouse. **Middle Row:** Runtime. **Bottom Row:** SoCs. LENS maintains high success rates across all structured maps. Runtime and SoCs are averaged over successful runs. Points are omitted when an algorithm produces no successful runs in that setting; thus, the SoCs is undefined rather than missing.

More strikingly, in the **City** environment (Table 4), removing the U-Net results in a **0**% success rate. This is not an indictment of the A* algorithm itself, but a confirmation of the synergy within our framework. While the ARG provides a strategic subgoal, a naive A* planner, seeking the shortest path to this intermediate point, is blind to global context. It guides agents into geographic traps (e.g., entering a U-shaped obstacle from the wrong side). In contrast, the U-Net's potential field is conditioned on the global goal vector via FiLM, producing globally-aware local proposals.

**Efficacy of the Defense Cascade:** The *Defense Cascade* proves essential for resolving dense, multi-agent conflicts (Table 4). Disabling the entire cascade reduces the success rate to 65.6%, indicating that purely reactive policies are insufficient for structured deadlocks. Breakdown of the cascade levels reveals: **L-CBS:** Removing Local CBS causes the sharpest drop (63.4%), confirming its role as the primary solver for local gridlocks. **Push-and-Rotate:** Removing the randomized heuristic also degrades performance (62.5%), highlighting its importance in breaking symmetric livelocks. Removing the **Endgame Solver** causes the success rate to plummet to 3.9%. Without this final global coordination, a few remaining stragglers often fail to reliably converge over vast distances, leading to episode timeouts.

## 6   Conclusion

LENS addresses the dichotomy between data-intensive foundation models and myopic reactive heuristics in Centralized Collaborative PO-MAPF. By decoupling topological guidance from conflict resolution, LENS employs a lightweight U-Net as a local heuristic estimator and relies on a bounded Defense Cascade to resolve structural deadlocks within localized spatio-temporal windows. This avoids the overhead of global replanning while providing bounded collision-free execution within each window. Future work will focus on:

Table 4: **Ablation Study.** *Left:* Component analysis in Direct Mode on Maze maps with 64 agents. *Right:* Component analysis in Incremental Discovery Mode on City maps ($64 \times 64$) with 128 agents.

| Direct Mode (Maze, 64 agents) | | | Incremental Discovery Mode (City, 128 agents) | | | |
|---|---|---|---|---|---|---|
| Configuration | SR | SoC (Runtime) | Configuration | SR | ISR | SoC |
| **LENS (Full)** | **78.7%** | 6117.55 (35.81s) | **LENS (Full)** | **77.1%** | **94.0%** | 12440.32 |
| *Impact of Learning:* | | | *Impact of Guidance:* | | | |
| w/o U-Net (using A*) | 64.1% | 3688.35 (21.06s) | w/o ARG | 35.9% | 82.9% | 8830.35 |
| *Impact of Defense Cascade:* | | | | | | |
| w/o Defense Cascade (Full) | 65.6% | 3608.46 (20.74s) | w/o Heuristic Input | 41.4% | 84.4% | 12051.43 |
| w/o Local CBS | 63.4% | 3830.66 (21.32s) | w/o Endgame Solver | 3.9% | 90.3% | 9635.60 |
| w/o Coordinated Retreat | 68.0% | 3609.08 (20.51s) | w/o U-Net (using A*) | 0.0% | 26.2% | N/A |
| w/o Push-and-Rotate | 62.5% | 3592.60 (20.35s) | w/o DEZD | 71.1% | 93.5% | 13354.87 |

(1) distributed regional solvers to reduce centralization; (2) socially-aware U-Net training for conflict-aware proposals; and (3) active exploration objectives to mitigate heuristic misguidance caused by unseen obstacles.

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

# A  Additional Evaluation: Puzzles Benchmark

To specifically test fine-grained coordination and deadlock resolution in highly constrained spaces, we evaluated LENS in Setting A on the **Puzzles** benchmark from POGEMA. These small, hand-crafted maps require multi-step, sliding-tile-puzzle-like maneuvers that defeat greedy policies.

As shown in Tables 5 and 6, LENS achieves a **100% Success Rate**, matching the centralized oracle (LaCAM) and the large-scale model (MAPF-GPT). Notably, our Sum of Costs (SoC) is significantly lower than MAPF-GPT's (e.g., 20.94 vs 97.43 for 4 agents), indicating that our L-CBS component finds much more efficient solutions in these tight logical puzzles than pure learned policies.

| Table 5: Puzzles Benchmark (4 agents) | | |
|---|---|---|
| **Configuration** | **SR (%)** | **Avg. SoC** |
| LENS | **100** | **20.94** |
| LaCAM | 100 | 28.68 |
| MAPF-GPT-85m | 100 | 97.43 |

| Table 6: Puzzles Benchmark (3 agents) | | |
|---|---|---|
| **Configuration** | **SR (%)** | **Avg. SoC** |
| LENS | **100** | **14.94** |
| LaCAM | 100 | 23.83 |
| MAPF-GPT-85m | 100 | 40.50 |

# B  Corridor Case Study: Quantifying Reactive Myopia

This benchmark operationalizes the effective planning horizon $H_\pi$ and directly measures the myopia claim of Contribution 1.

**Setup.** We construct four corridor-dominant maps with corridor depths in $\{5, 6, 7, 8\}$, so that a $H_\pi{=}1$ planner cannot distinguish a dead-end entry from a through-passage before committing. Figure 4 shows two representative layouts with agents' start positions (filled) and goals (hollow) marked along and across corridors, forcing head-on encounters inside narrow passages. All methods run under Setting B with the same $\mathcal{M}_{shared}$: **CS-PIBT** ($H_\pi{=}1$), **LNS2-RL** (global LNS with learned local repair, heuristic recomputed on map updates), **LENS** ($H_\pi{\leq}10$). Each configuration is evaluated over 4 maps with multiple seeds.

Table 7: **Corridor Case Study.** Performance under Setting B on four corridor-dominant maps at two agent densities. SoC and Makespan are computed strictly over successful runs (survivorship bias caveat applies: methods with low SR average only over their easiest instances). Best SR per density in **bold**.

| Method | 8 agents | | | | | 16 agents | | | | |
|---|---|---|---|---|---|---|---|---|---|---|
| | SR | ISR | SoC | Mksp. | Time (s) | SR | ISR | SoC | Mksp. | Time (s) |
| CS-PIBT ($H_\pi{=}1$) | 0.95 | 0.988 | **167.42** | **28.58** | **0.18** | 0.200 | 0.603 | **495.75** | **48.25** | **0.75** |
| LNS2-RL (local repair) | 0.50 | 0.700 | 440.50 | 88.80 | 13.60 | 0.000 | 0.272 | – | – | 19.23 |
| **LENS** ($H_\pi{\leq}10$) | **1.000** | **1.00** | 478.50 | 74.00 | 1.19 | **0.562** | **0.696** | 1439.00 | 146.56 | 14.91 |

**Horizon effect.** Doubling density produces three distinct collapse patterns. CS-PIBT's 1-step shield cannot foresee that entering a narrow corridor leads to bidirectional deadlock and oscillates once trapped. LNS2-RL's global search reasons over long horizons, but each map update forces heuristic recomputation, and its local repair operator re-proposes moves into the same trap. LENS is the only method retaining >50% SR at 16 agents, exceeding CS-PIBT by 36.2 pp and LNS2-RL by 56.2 pp under identical map information—attributable to horizon, not to map awareness.

**ISR signature of myopia.** At 16 agents, CS-PIBT's ISR=0.603 against SR=0.200 indicates that most individual agents finish but at least one remains corridor-locked per map—the signature of 1-step myopia.

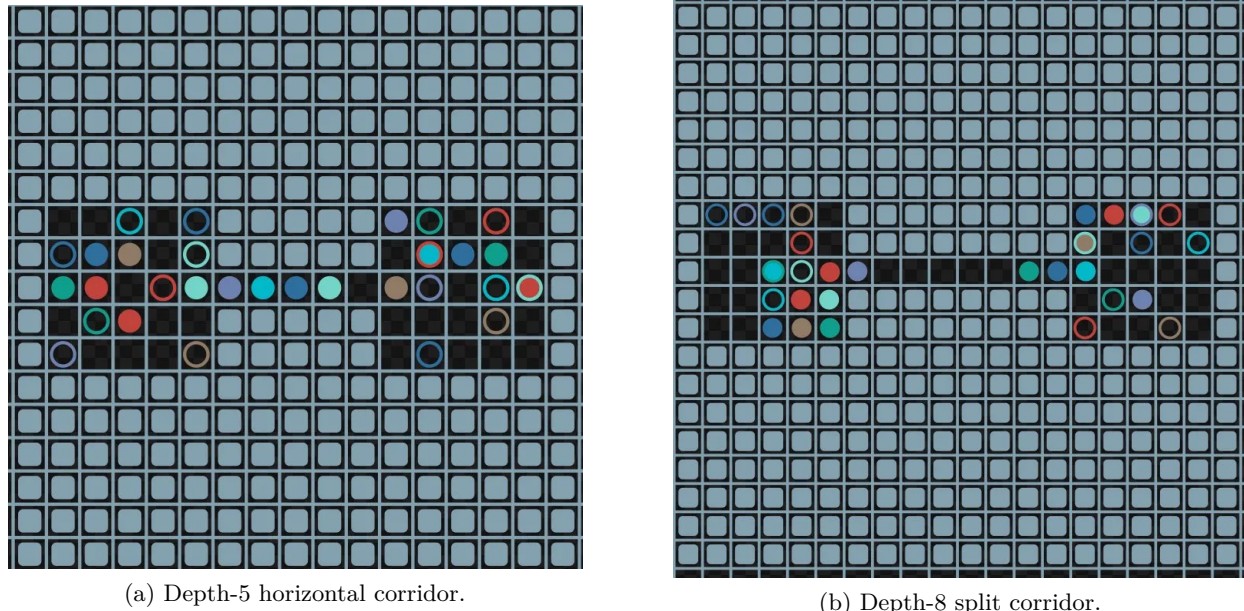

(a) Depth-5 horizontal corridor.

(b) Depth-8 split corridor.

Figure 4: **Representative corridor maps.** Filled circles denote agent start positions; hollow circles denote goals. Start–goal assignments are arranged to force head-on encounters inside corridors, where traversal cost exceeds $H_\pi=1$.

LENS's tighter ISR-to-SR gap (0.696 vs. 0.562) reflects the Defense Cascade resolving conflicts before they become full-system deadlocks.

**Cost of caution.** At 8 agents, LENS yields higher SoC than CS-PIBT (478.5 vs. 167.4). Part of this is survivorship bias (CS-PIBT averages only over its easier 95% subset); part is genuine multi-hop descent takes conservative detours. This is the price paid for the robustness that prevents collapse at 16 agents.

## C    Limitations

While LENS demonstrates strong performance, we acknowledge limitations:

- **Centralization Bottleneck:** The Defense Cascade is centralized. Although triggered locally for small groups, extreme congestion merging multiple groups could create a computational bottleneck requiring reliable communication.

- **Dependence on Exploration:** The search-based resolver relies on the shared global map $\mathcal{M}_{shared}$. If critical paths remain unexplored (marked UNKNOWN), the resolver may fail even if a path exists.

- **OOD Generalization:** The U-Net proposer, trained on smaller maps, may struggle with unseen complex obstacle textures in OOD maps (e.g., Cities), necessitating more frequent cascade interventions.

## D    Computational Profiling and Defense Cascade Statistics

To transparently analyze the runtime characteristics and the internal behavior of the LENS framework, we present a granular computational profile based on our experiments in completely unknown environments (Setting B).

**Defense Cascade Utilization.** To validate the necessity and efficiency of our hierarchical conflict resolution, we logged the invocation frequency of each defense layer in highly constrained topologies (Maze

maps with 64 agents). As shown in Table 8, out of 13,415 encountered conflict groups, the standard Local CBS (L1) successfully resolved 98.96% of them. This demonstrates that local search is highly capable of untangling congestion when guided by topology-aware potential fields. The heuristic fallbacks (L2-L4) are rarely invoked ($\approx 1\%$), confirming their role as strict extreme-case safety nets rather than primary planners.

Table 8: Defense Cascade Usage Frequency (Maze, 64 Agents, Setting B)

| Defense Layer | Successful Invocations | Proportion (%) |
|---|---|---|
| L1: Standard CBS (L-CBS) | 13,276 | 98.96% |
| L2: Coordinated Retreat | 72 | 0.54% |
| L3: Push-and-Rotate | 66 | 0.50% |
| **Total Conflict Groups** | **13,415** | **100.00%** |

**Latency Breakdown in Massive Environments.** In the City-scale maps (256 agents), LENS incurs a higher average runtime (184.04s) compared to purely reactive baselines. To identify the bottleneck, we profiled the per-step latency (Table 9). The analysis reveals that the core algorithmic components of LENS are extremely lightweight: the U-Net neural inference accounts for only 1.18% of the total time, and the L-CBS conflict resolution consumes merely 3.66%. The vast majority of the computational load (78.82%) is dedicated to the continuous updating of the global map ($\mathcal{M}_{shared}$) and pattern detection across 256 agents. This confirms that the observed runtime overhead in massive maps is an engineering artifact of centralized state synchronization, rather than a scalability limit of the hybrid learning-search architecture itself.

Table 9: Computational Latency Profile (City Map, 256 Agents, Setting B)

| Pipeline Component | Total Time (s) | Proportion (%) |
|---|---|---|
| 1. Map Update | 148.26 | 70.11% |
| 2. High-Level Planning and ARG Guidance | 50.93 | 24.08% |
| 3. Conflict Resolution (L-CBS) | 6.88 | 3.25% |
| 4. U-Net Inference | 2.82 | 1.33% |
| 5. Path Execution | 2.58 | 1.22% |

# E   Baselines and Adaptation Details

To comprehensively evaluate the LENS framework across varying degrees of environmental uncertainty, we compared it against five state-of-the-art multi-agent pathfinding algorithms representing distinct paradigms: classical centralized search, communication-based MARL, hybrid search-learning, reactive heuristics, and large-scale foundation models. This section details their specific training configurations and the engineering adaptations required for execution in strictly unknown environments (Setting B).

## E.1   Baseline Descriptions and Training Configurations

- **LaCAM (Okumura, 2023):** Serving as our centralized search oracle, LaCAM represents the state-of-the-art in sub-optimal, complete MAPF solvers. It operates with perfect global knowledge and pre-computed global heuristics, providing a theoretical upper bound for solution quality (Sum of Costs) and success rates in fully observable settings (Setting A).

- **SCRIMP (Wang et al., 2023):** A state-of-the-art multi-agent reinforcement learning (MARL) planner that leverages Transformer-based communication mechanisms for decentralized coordination and local tie-breaking among agents.

- **LNS2-RL (Li et al., 2022a):** A hybrid solver that integrates a MARL policy into the Large Neighborhood Search (LNS) framework to repair local conflicts dynamically. *Training Setup:* The underlying neural policy was trained via reinforcement learning exclusively on *Random* maps with varying grid sizes (from $10 \times 10$ to $50 \times 50$) and obstacle densities ranging from 0% to 50%.

- **CS-PIBT (Veerapaneni et al., 2025):** A robust reactive decentralized heuristic that combines a learned Safe Sequential Imitation Learning (SSIL) policy with a strict collision-avoidance shield (PIBT). *Training Setup:* Unlike methods trained solely on random grids, the SSIL policy utilizes a diverse dataset derived directly from the *MovingAI benchmark*, exposing the model to highly structured topologies such as cities and warehouses during training.

- **MAPF-GPT (Andreychuk et al., 2025b):** A large-scale neural foundation model designed to imitate centralized optimal planners entirely end-to-end. *Training Setup:* The model was pre-trained on a massive offline dataset comprising over *1 billion* expert observation-action pairs. To encourage long-horizon reasoning, the dataset distribution is heavily skewed toward topologically complex structures, consisting of 50% *Maze-like* maps and 50% *Random* maps.

### E.2 Adaptations for Incremental Map Discovery (Setting B)

Setting B simulates realistic robotic deployments where the global map is initially hidden. Agents must navigate using limited local fields of view while synchronously updating a shared persistent map ($\mathcal{M}_{shared}$). To ensure a rigorous and fair comparison, we adapted the applicable baselines to interface with this incremental discovery framework:

- **SCRIMP:** As a decentralized MARL policy operating natively on local observations, SCRIMP requires no structural adaptation and is naturally suited for online execution in Setting B.

- **Adapted LNS2-RL:** The algorithm is granted real-time access to the dynamically updating $\mathcal{M}_{shared}$. However, because its neighborhood search fundamentally relies on a global heuristic matrix, we enforce a strict recomputation protocol: whenever an agent's FOV reveals a new obstacle, LNS2-RL must recompute its global heuristic graph from scratch before proposing the next joint action. This adaptation preserves its functionality but inevitably incurs severe computational overhead in dense maps.

- **Adapted CS-PIBT:** Similarly, CS-PIBT is granted read-access to $\mathcal{M}_{shared}$. Its guiding heuristic graph is dynamically recomputed in real-time upon any obstacle updates registered by the swarm, allowing the SSIL policy and PIBT shield to operate over the most current topological belief.

**Excluded Methods in Setting B:** Certain state-of-the-art methods were fundamentally incompatible with the strictly unknown environment constraint and were therefore excluded from Setting B. **MAPF-GPT** (Andreychuk et al., 2025b) relies intrinsically on pre-computed global cost-to-go matrices as inputs to its Transformer backbone; it cannot operate under online map discovery without a complete architectural redesign. Similarly, **DMCTS** (Skrynnik et al., 2024b)—despite explicitly handling peer uncertainty—utilizes a neural policy (COSTTRACER) that requires a pre-computed distance matrix over a fully known, static global graph. This assumption violates the constraints of Setting B and cannot be straightforwardly resolved via online heuristic recomputation without breaking the policy's theoretical guarantees.

## F  Implementation Details

Our LENS framework is implemented in Python, leveraging several key open-source libraries to ensure efficiency and robustness.

**Core Technologies** The Neural Proposer, our U-Net model, is built using **PyTorch**. The environment for both expert trajectory generation and final evaluation is the **POGEMA** benchmark, which provides a flexible and standardized platform for MAPF research.

**Algorithmic Components.** The classical search components of our framework rely on well-established libraries. Graph-based operations, such as conflict group identification and the high-level planning in the ARG for LENS-H, are implemented using **NetworkX**. To accelerate computationally intensive, low-level functions such as heuristic calculations, we employ **Numba** for Just-In-Time (JIT) compilation, which significantly reduces the overhead of these frequently called routines. The core conflict resolution solver, local CBS, is a custom implementation optimized for small agent groups and limited planning horizons.

## F.1  Neural Proposer Architecture

The Neural Proposer is implemented as a modified U-Net with the following specifications:

**Inputs:**

- **Spatial Tensor ($4 \times L \times L$):** Channel 0: Static Obstacles; Channel 1: Dynamic Agents; Channel 2: Ego-Agent Position; Channel 3: Projected Local Target (the subgoal coordinate or local guiding path).

- **Non-Spatial Vector ($2 \times 1$):** Normalized vector $(\Delta x, \Delta y)$ pointing to the current subgoal.

**Architecture:**

- **Encoder:** 3 blocks of `DoubleConv` (Conv3x3 $\rightarrow$ BN $\rightarrow$ ReLU $\rightarrow$ Conv3x3 $\rightarrow$ BN $\rightarrow$ ReLU $\rightarrow$ **SEBlock**). Downsampling via MaxPool2d. Channels: $64 \rightarrow 128 \rightarrow 256 \rightarrow 512$.

- **Bottleneck & FiLM:** The bottleneck features are modulated by the **FiLM Layer**. The non-spatial vector is projected via an MLP to generate scale ($\gamma$) and shift ($\beta$) parameters.

- **Decoder:** 3 blocks of Transposed Convolution for upsampling followed by `DoubleConv`. Skip connections concatenate encoder features.

- **Output Head:** $1 \times 1$ Convolution to produce the $1 \times L \times L$ potential field.

# G  Data Generation and Training

## G.1  Dataset Technical Details

To ensure our work is comparable to existing state-of-the-art models, we adopted a data generation strategy similar to that of MAPF-GPT, focusing on the same types of environments. Our dataset was generated exclusively on the **Mazes** and **Random** map types from the POGEMA benchmark (Table 10).

While MAPF-GPT generated over a billion samples, our targeted sampling strategy (focusing on initialization and terminal approach phases) resulted in a compact dataset of approximately **1.3 million samples**. This highlights the data efficiency of our hybrid learning approach compared to pure end-to-end imitation learning.

## G.2  Visualization of U-Net Inputs and Outputs

The input and output spaces of the U-Net are visualized in Figure 5 to clarify the training sample generation process. This learning task is framed as *Heuristic Approximation*, where the U-Net is trained to regress a dense local crop of the optimal global potential field rather than predicting discrete actions. The global context, shown on the left, is derived offline via backward BFS from the target goal on a fully known map, with colors representing the raw distance to the destination.

During online execution, the U-Net utilizes an $11 \times 11$ egocentric spatial tensor consisting of multiple channels. The visible spatial inputs include binary occupancy grids for *Obstacles* and *Other Agents*. Although not

Table 10: Details of map sets from the POGEMA benchmark used for training and evaluation.

| Set | Agents | Maps | Map Size | Seeds | Steps |
|---|---|---|---|---|---|
| Random / Mazes | 8,16,32,48,64 | 128 | $17 \times 17$–$21 \times 21$ | 1 | 128 |
| Warehouse | 32,64,128,156,192 | 1 | $33 \times 46$ | 128 | 128 |
| Cities-tiles | 64,128,192,256 | 128 | $64 \times 64$ | 1 | 512 |
| Puzzles | 2, 3, 4 | 16 | $5 \times 5$ | 10 | 128 |

shown as a separate patch in the figure, the self-position(channel 2) and *Projected Local Target* (Channel 3) employ a high-intensity representation analogous to the *Other Agents* channel. When the global goal is within the FOV, this channel renders a single high-intensity coordinate; when the goal is distant, it encodes the hierarchical guidance path from the ARG as a decaying signal along the projected trajectory. These spatial features are fused with a normalized *Direction to Goals* vector $(\Delta x, \Delta y)$ via a FiLM layer at the bottleneck. The final *Prediction Label*, shown at the bottom right, represents the corresponding local patch of the potential field, normalized to a $[0, 10]$ scale to facilitate stable gradient learning and efficient path extraction through greedy descent.

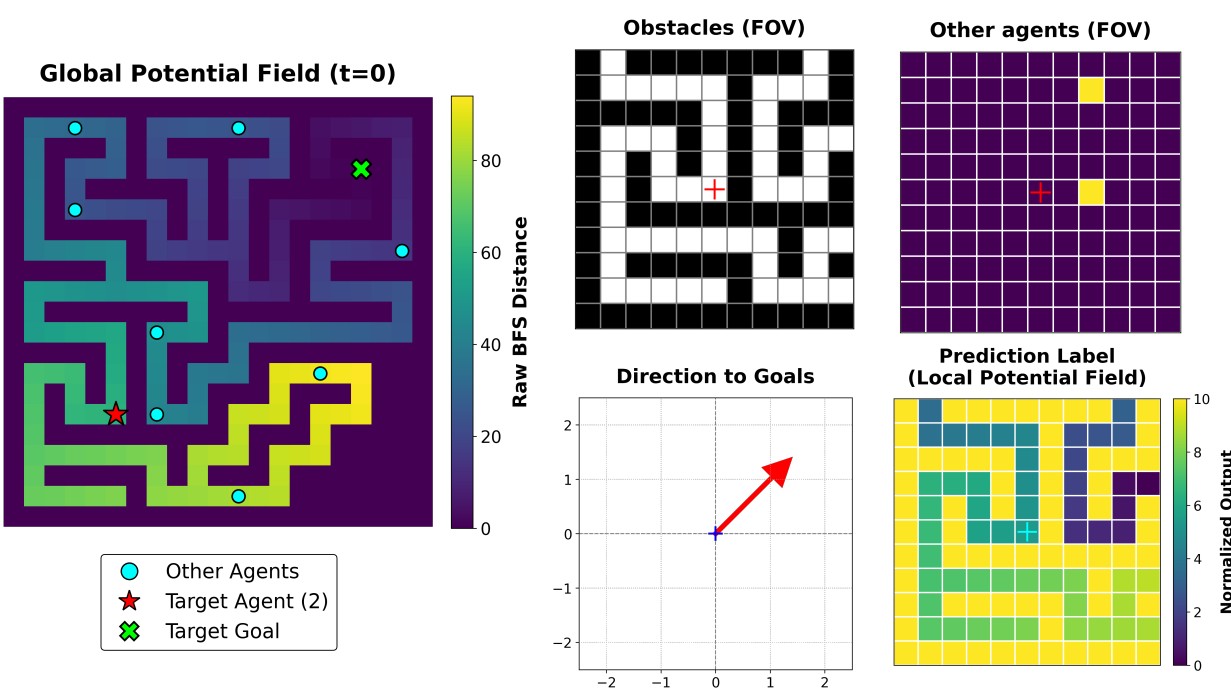

Figure 5: Data Creation and Input/Output Visualization for the Neural Proposer. The global potential field (left) is used strictly offline to generate the cropped *Prediction Label* (bottom right). During online inference, the U-Net utilizes only the local FOV channels and the goal direction vector to predict the local potential field. The self-position(channel 2) and *Projected Local Target* (Channel 3) are not shown here.

### G.3 Training Hyperparameters and Stability

The training was performed on a single NVIDIA A40 GPU. The key hyperparameters are detailed in Table 11. To demonstrate stability, Figure 7 illustrates the loss curves. The model converged smoothly, and training was halted early at epoch 93 via a patience mechanism, selecting the best model checkpoint at epoch 83 (validation loss = 0.2199).

Table 11: Key hyperparameters for U-Net training.

| Parameter | Value |
|---|---|
| *Training Setup* | |
| Epochs | 100 |
| Batch Size | 1024 |
| Initial Learning Rate | 5e-4 |
| Optimizer | Adam |
| LR Scheduler | ReduceLROnPlateau (Patience 5, Factor 0.1) |
| Early Stopping Patience | 10 epochs |
| *Loss Function* | |
| Primary Loss | MSE |
| Auxiliary Loss | Gradient L1 Loss ($\lambda_{grad} = 0.1$) |

### G.4 Impact of Partial Observability on Heuristic Accuracy

To quantitatively investigate how unseen obstacles affect the Neural Proposer's ability to approximate the global potential field, we conducted an isolated evaluation of the U-Net's prediction error under varying levels of environmental uncertainty.

**Experimental Setup**   We ran simulations using 2 agents on Cities-tiles maps. We define the *Proportion of Unseen Obstacles* as the ratio of true static obstacles that currently remain marked as "Unknown" within the agent's persistent map ($\mathcal{M}_{shared}$) relative to the total number of true obstacles in the entire environment. To rigorously evaluate the accuracy of the neural prediction, we compute the **Mean Squared Error (MSE)** between the U-Net's predicted local potential field and the ground-truth potential field (derived from a global BFS on the fully known map, cropped to the agent's FOV).

**Findings and System Robustness**   As illustrated in Figure 6, regression analysis demonstrates a clear **positive correlation** between the proportion of unseen obstacles and the Normalized MSE. When critical geometric structures (such as the boundaries of a U-shaped trap) remain unobserved and outside the agent's $11 \times 11$ field of view, the Neural Proposer is forced to make "optimistic" predictions. This leads to a distorted local gradient that may temporarily misguide the agent toward dead ends. This result quantitatively validates the inherent vulnerability of purely reactive neural heuristics in strictly unknown environments.

### G.5 Qualitative Analysis of Neural Potential Fields

To validate that our Neural Proposer learns to infer global topology from local observations rather than merely memorizing maps, we visualize the predicted potential fields in distinct navigation scenarios. Figure 8 compares the local input, the ground-truth potential field (derived from global BFS), and the model's prediction.

As demonstrated in the **Dead End (Trap)** scenario (Middle Row), a naive Euclidean heuristic would assign a high value to the position inside the U-shape due to its proximity to the goal. However, our model, trained on global ground-truth potentials, correctly predicts a shadow of high cost (represented by

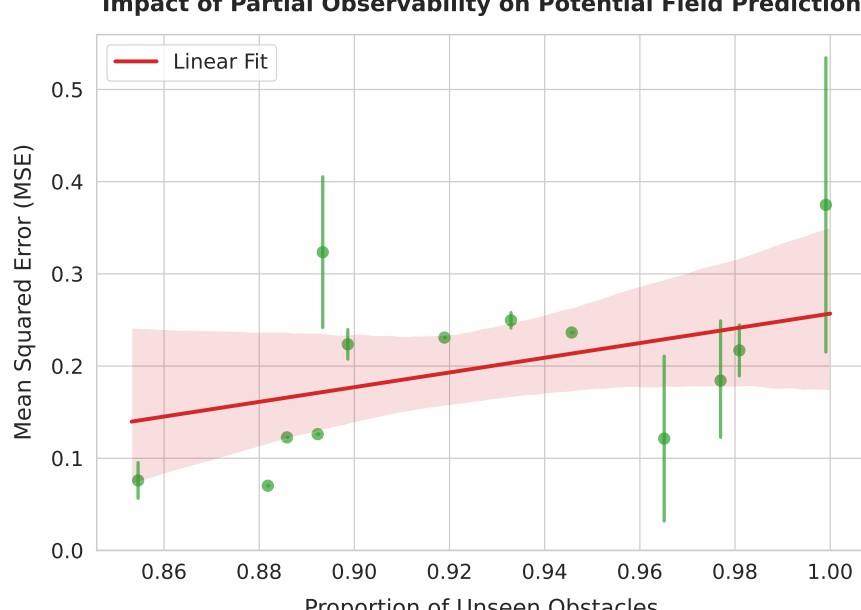

Figure 6: Correlation between the proportion of unseen obstacles and the Normalized Mean Squared Error (MSE) of the Neural Proposer's local potential field. The red linear fit with a 95% confidence interval demonstrates that as environmental uncertainty increases, the topographical distortion of the learned heuristic also increases, validating the "optimistic" bias under partial observability.

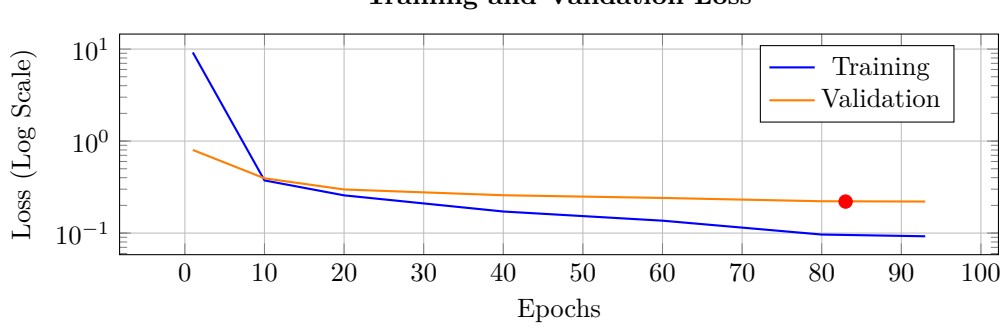

Figure 7: Training/Validation Loss. Training stopped early at epoch 93.

dark pixels) inside the trap. This visualization confirms that the U-Net successfully reconstructs non-local constraints—recognizing that the path is blocked beyond the visual horizon—thereby enabling the agent to avoid structural local minima that plague reactive baselines like CS-PIBT.

## H Defense Cascade Algorithms

This appendix provides the detailed algorithmic implementation of the multi-level Defense Cascade used in LENS. The cascade is triggered when the Deadlock Entry Zone Detector (DEZD) identifies agents in a non-progressive state.

### H.1 Definitions and Notation

For clarity and consistency across the algorithms, we define the following mathematical notation:

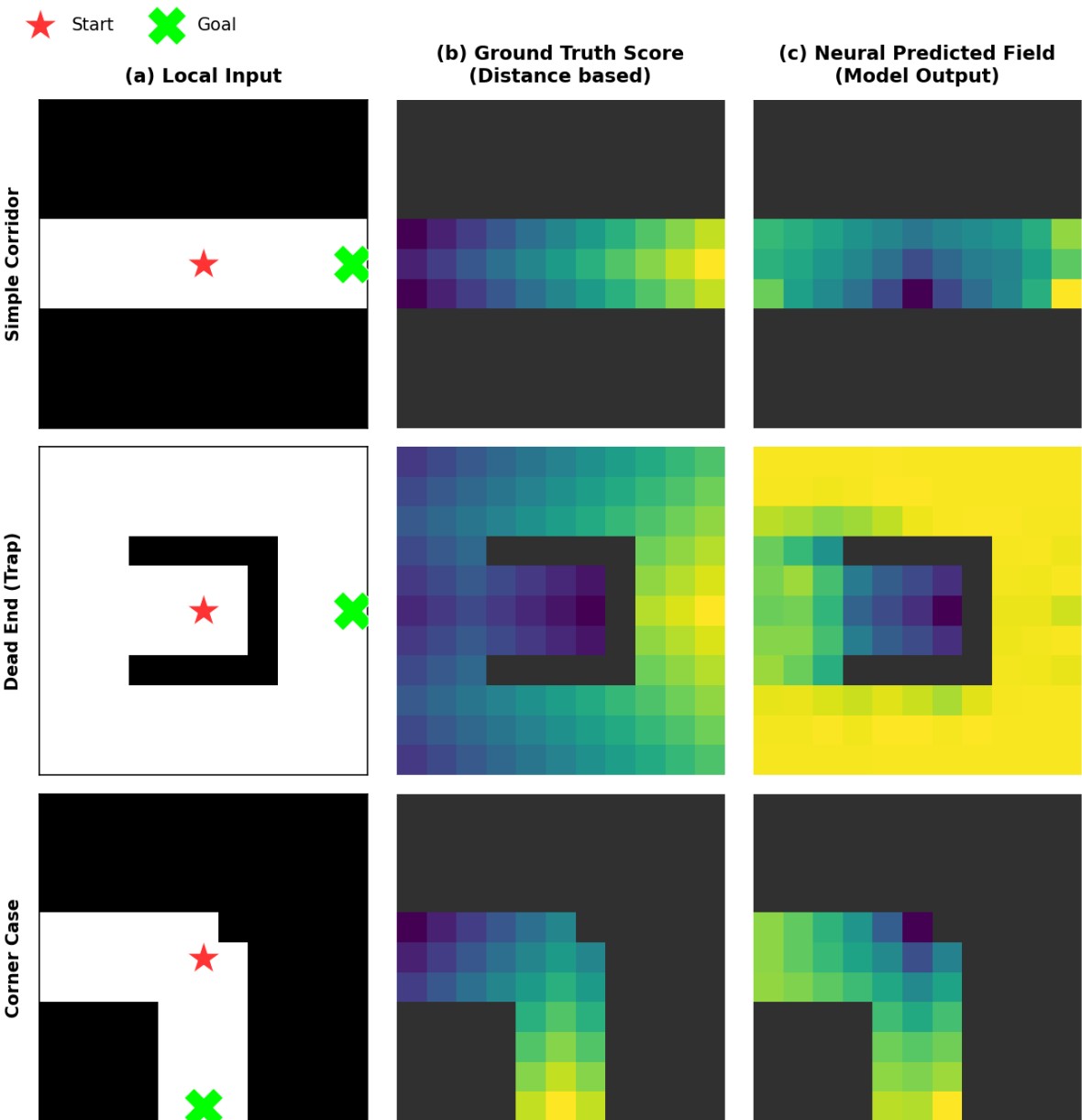

Figure 8: **Qualitative Comparison of Learned Potential Fields.** We visualize the Local Input (Left), the Global Ground Truth derived from BFS (Center), and the U-Net's Predicted Field (Right). **Top Row (Corridor):** The model correctly predicts a smooth gradient along the clear path. **Middle Row (Dead End/Trap):** This is the critical case. Although the goal (Green X) is geometrically close to the agent (Red Star), it is topologically distant due to the U-shaped wall. The Ground Truth reflects this with low scores (dark colors) inside the trap. Crucially, the Neural Proposer *reproduces this topological awareness*, predicting a low-value field inside the trap, effectively inhibiting the agent from entering the deadlock. **Bottom Row (Corner):** The model successfully infers the turning geometry.

## H.2 Level 1: Localized Conflict-Based Search (L-CBS)

L-CBS is the primary execution kernel activated upon conflict detection. As implemented in our codebase, it differs from standard CBS in two critical ways: it is restricted to a small spacetime window, and its low-level search is actively guided by the neural potential field.

Table 12: Mathematical Notation for Defense Cascade Algorithms

| Symbol | Description |
|---|---|
| $\mathcal{I}_{conf}$ | A subset of agents identified as being in a conflict group or deadlock. |
| $n_i$ | An individual agent with index $i$. |
| $s_t^i = (x, y)$ | The position of agent $n_i$ at time step $t$. |
| $g^i$ | The global goal position of agent $n_i$. |
| $M_{local}$ | The local grid map representing static obstacles within the relevant area. |
| $\hat{\Phi}^i$ | The learned local potential field for agent $n_i$ (output of the U-Net). |
| $\mathcal{N}(s)$ | The set of valid (traversable) neighboring cells of position $s$. |
| $h_{dist}(s_1, s_2)$ | Heuristic distance function (e.g., Manhattan distance) between two points. |
| $\mathcal{W}$ | A set of spacetime constraints for CBS (e.g., $(n_i, s, t)$ means $n_i$ cannot be at $s$ at time $t$). |
| $k$ | The fixed short time horizon for the localized search (e.g., $\min(d_{\text{subgoal}}, 10)$ steps). |
| $\delta_{pad}$ | Spatial padding added to the bounding box of conflict agents. |

---

**Algorithm 1** Level 1: Localized Conflict-Based Search (L-CBS)

---

1: **Input:** Conflict agents $\mathcal{I}_{conf}$, local map $M_{local}$, potential fields $\{\hat{\Phi}^i\}_{i \in \mathcal{I}_{conf}}$, horizon $k$, padding $\delta_{pad}$.
2: **Output:** Collision-free path segments $\mathcal{P} = \{P^i\}$ for $t \in [0, k]$ or FAILURE.
3: // **Phase 1: Define Search Scope**
4: Calculate spatial bounding box defined by $\{s_0^i\}$ and goals $\{g^i\}$ for all $n_i \in \mathcal{I}_{conf}$.
5: Define localized region $R$ by expanding the bounding box by $\delta_{pad}$.
6: Construct temporary map $M_R$ by treating all cells outside $R$ as obstacles.
7: // **Phase 2: High-Level CBS**
8: $Root.\mathcal{W} \leftarrow \emptyset$
9: **for** each agent $n_i \in \mathcal{I}_{conf}$ **do**
10:     $Root.paths[i] \leftarrow$ LowLevelSearch$(n_i, s_0^i, M_R, \emptyset, \hat{\Phi}^i)$
11: **end for**
12: $Root.cost \leftarrow \sum Cost(Root.paths[i])$
13: $OPEN \leftarrow \{Root\}$
14: **while** $OPEN$ is not empty **do**
15:     $N \leftarrow \arg\min_{Node \in OPEN} Node.cost$
16:     Pop $N$ from $OPEN$.
17:     // **Validate paths for vertex/edge conflicts within horizon** $k$
18:     $C \leftarrow$ FindFirstConflict$(N.paths[0 \ldots k])$
19:     **if** $C$ is None **then**
20:         **return** $N.paths$ {Optimal local solution found}
21:     **end if**
22:     // **Branching on conflict** $C = (n_i, n_j, s, t)$
23:     **for** each agent $n_k \in \{n_i, n_j\}$ **do**
24:         $NewNode \leftarrow$ copy of $N$
25:         Add constraint $(n_k, s, t)$ to $NewNode.\mathcal{W}$
26:         // **Re-plan with potential-field guidance**
27:         $NewNode.paths[k] \leftarrow$ LowLevelSearch$(n_k, s_0^k, M_R, NewNode.\mathcal{W}, \hat{\Phi}^k)$
28:         **if** $NewNode.paths[k]$ found **then**
29:             Update $NewNode.cost$ and insert into $OPEN$.
30:         **end if**
31:     **end for**
32: **end while**
33: **return** FAILURE {Trigger Level 2}

---

34: **Procedure** LowLevelSearch$(n_i, start, map, \mathcal{W}, \hat{\Phi}^i)$
35: Run Space-Time A* on $map$ subject to $\mathcal{W}$.
36: **Crucially**, define the heuristic cost $h(s)$ for state $s$ using the neural potential field:
37: $h(s) = \hat{\Phi}^i(s)$ {Leverage learned topological guidance}
38: **return** shortest path or None

---

### H.3 Level 2: Coordinated Retreat

This module is triggered when L-CBS fails, typically due to structural deadlocks (e.g., a blocked narrow corridor) where no forward progress is possible within the search horizon. It resolves conflicts by explicitly breaking symmetry using a Yielder/Mover paradigm based on topological progress.

---
**Algorithm 2** Level 2: Coordinated Retreat
---
1: **Input:** Conflict agents $\mathcal{I}_{conf}$, local map $M_{local}$, global goals $\{g^i\}$.
2: **Output:** Retreat paths for yielders, hold commands for movers.
3: **// Phase 1: Classification**
4: For each $n_i \in \mathcal{I}_{conf}$, calculate progress score $S_i = h_{dist}(s_t^i, g^i)$.
5: Sort $\mathcal{I}_{conf}$ in descending order of $S_i$ (furthest from goal first).
6: Partition $\mathcal{I}_{conf}$ into $\mathcal{I}_{yielder}$ (top 50%) and $\mathcal{I}_{mover}$ (bottom 50%).
7: **// Phase 2: Yielder Planning**
8: Initialize plans $\mathcal{P} \leftarrow \emptyset$.
9: **for** each yielder $n_y \in \mathcal{I}_{yielder}$ **do**
10:    Determine retreat direction vector $\vec{d}_{retreat} = -1 \cdot \text{normalize}(g^y - s_t^y)$.
11:    Perform BFS starting from $s_t^y$ in direction roughly matching $\vec{d}_{retreat}$ on $M_{local}$.
12:    Search for closest "junction" state $s_{junc}$ (defined as a cell with $> 2$ free neighbors or an open area).
13:    **if** $s_{junc}$ found **then**
14:       Plan path $P^y$ from $s_t^y$ to $s_{junc}$ treating other agents as static obstacles.
15:       Add $P^y$ to $\mathcal{P}$.
16:    **else**
17:       Add hold position command for $n_y$ to $\mathcal{P}$.
18:    **end if**
19: **end for**
20: **// Phase 3: Mover Planning**
21: **for** each mover $n_m \in \mathcal{I}_{mover}$ **do**
22:    Add hold position command for $n_m$ to $\mathcal{P}$ (waiting for yielders to clear).
23: **end for**
24: **return** $\mathcal{P}$
---

### H.4 Level 3: Heuristic Agitation

This is the final fallback for tightly interlocked configurations (e.g., symmetric $2 \times 2$ cycles) where neither structured search nor retreat logic applies. It applies a randomized perturbation to break the symmetry of the deadlock.

---
**Algorithm 3** Level 3: Heuristic Agitation (Push-and-Rotate)
---
1: **Input:** Conflict agents $\mathcal{I}_{conf}$, local map $M_{local}$.
2: **Output:** Single-step actions for $\mathcal{I}_{conf}$.
3: Assign random priorities to agents in $\mathcal{I}_{conf}$.
4: **for** each agent $n_i \in \mathcal{I}_{conf}$ in priority order **do**
5:    Identify set of valid unoccupied neighbors: $V_i = \{s' \in \mathcal{N}(s_t^i) \mid s' \text{ is free in } M_{local} \text{ and unoccupied}\}$.
6:    **if** $V_i$ is not empty **then**
7:       Select a random neighbor $s_{next} \in V_i$.
8:       Assign action: move to $s_{next}$.
9:       Mark $s_{next}$ as occupied for subsequent agents in this step.
10:    **else**
11:       Assign action: hold position ($s_t^i$).
12:    **end if**
13: **end for**
14: **return** Assigned actions.
---

### H.5   Level 4: Endgame Solver

In massive scenarios (e.g., Cities), standard collision evaluation may time out if disconnected stragglers indefinitely fail to converge. The Endgame Solver monitors the active agent density. Once this density drops below a critical threshold (e.g., 15%), it explicitly disables localized spatio-temporal windows and transitions to a globally bounded CBS. This solver searches over the full $\mathcal{M}_{shared}$ with an extended iteration limit to guarantee the final sparse conflicts are optimally resolved, ensuring episode termination.

## I   System Hyperparameters

Table 13: Inference and System Hyperparameters.

| Parameter | Value |
|---|---|
| *Environment* | |
| FOV Size ($L$) | $11 \times 11$ (Radius 5) |
| Max Episode Steps | 1024 |
| Timeout | 1000s |
| *Adaptive Region Graph (ARG)* | |
| Region Size | $16 \times 16$ |
| Update Frequency | Dynamic (on map discovery) |
| *Defense Cascade* | |
| Planning Horizon ($k$) | $\min(d_{\text{subgoal}}, 10)$ steps |
| Execution Window ($n$) | 3(small map) or 6(large map) steps |
| L-CBS Time Limit | 30s |
| L-CBS Max Iterations | 100 |
| DEZD Stagnation Window | 10 steps |
| Stagnation BBox Size | 8 cells |
| Evaporation Threshold | $> 12$ agents |

## J   Future Work

Future research will focus on: (1) **Distributed Resolution:** Evolving the Defense Cascade into a hierarchical system where regional solvers handle local conflicts to improve scalability. (2) **Socially Aware Learning:** Training the U-Net with soft penalties based on predicted paths of other agents to generate proposals that are inherently less conflict-prone. (3) **Active Exploration & Long-Horizon Attention:** Addressing the inherent limitation where unseen obstacles misguide local heuristics by incorporating uncertainty maps or entropy-reduction objectives into the U-Net's inputs, and augmenting the proposer with attention mechanisms to improve reasoning over larger spatial scales in real city-sized maps.

