# OpenReview forum: "LENS: Learning to Navigate with Active Search for Partially Observable MAPF in Unknown Environments"
_TMLR — Accepted by TMLR_

### Review · Reviewer_TZkQ · 2026-04-08

**Summary Of Contributions:**

This paper studies partially observable MAPF in initially unknown environments under an online centralized collaborative setting and proposes LENS, a hierarchical hybrid architecture with three coupled modules:
(i) high-level guidance from a synchronized shared explored map via an Adaptive Region Graph (ARG),
(ii) a Neural Proposer that predicts a dense local potential field from a local FOV plus guidance signal, and
(iii) a Defense Cascade for explicit conflict or deadlock repair, centered on bounded local CBS over forecast conflict groups, with retreat / push-and-rotate / endgame fallbacks. The key design choice is to amortize local topo-metric inference into a learned field while keeping discrete multi-agent legality explicit.

Technically, the learned component is well chosen. Rather than directly imitating actions, the proposer regresses a cropped global BFS potential field with both value and gradient-matching supervision, so the local descent structure is preserved. The proposer is therefore better interpreted as a learned local planner surrogate or heuristic field estimator than as a reactive policy.

Empirically, the paper shows that this proposal–repair decomposition is effective in structured, deadlock-heavy settings and degrades less than reactive baselines when moving from globally known maps (Setting A) to initially unknown maps with online exploration (Setting B). The ablations are useful, and the appendix profiling strengthens the architectural story: in the profiled maze regime, standard local CBS resolves most conflict groups, indicating that the main repair primitive, not the fallback heuristics, is doing nearly all of the real work.

The main strengths are:
- a technically coherent proposal–repair factorization;
- a learned target with stronger geometric structure than direct action imitation;
- experiments targeted at the claimed failure regime (structured unknown maps);
- and evidence that ARG, proposer, and local CBS all contribute materially.

The main weaknesses are:
- the paper’s novelty claims should be narrower relative to prior hybrid MAPF methods, learned local graph-routing methods, and deep-guided search methods;
- the framing of learned baselines is somewhat too coarse, especially given recent work suggesting that some dense-MAPF failures arise from pairwise interaction bottlenecks or attention dilution;
- the method is studied in a materially easier centralized collaborative regime with synchronized shared mapping;
- and some experimental claims, especially around robustness and efficiency, are somewhat stronger than what the current evidence fully isolates.

Overall, the paper is best positioned not as broadly pioneering learning with search for MAPF, but as introducing a specific and meaningful hybrid interface: learn a local topological proposal object and invoke explicit bounded repair only where the predicted interaction structure demands it.

**Audience:**

No

**Audience Explanation:**

Yes.

I think the paper would interest several groups within TMLR’s audience.

First, it should interest researchers in machine learning for combinatorial optimization / learning-guided planning and search, because it studies a central design question: what part of a structured decision problem should be learned, and what part should remain explicit? The paper gives a concrete answer in MAPF by learning the local topo-metric proposal while keeping hard multi-agent legality explicit.

Second, it should interest researchers in multi-agent learning and coordination, because it explores a nontrivial middle ground between end-to-end learned coordination and fully symbolic planning. The contribution is not just a benchmark result, but an architectural interface for mixing learned local intent with explicit group-local repair.

Third, it is directly relevant to the MAPF or multi-robot coordination community, especially for readers interested in partial observability, unknown environments, and structured deadlock regimes.

**Broader Impact Concerns:**

I do not have major broader-impact concerns that would materially affect the recommendation.

Overall, I do not see serious unresolved ethical concerns, but a brief acknowledgment of deployment assumptions and possible failure modes would be beneficial.

**Claims And Evidence:**

Yes

**Claims Explanation:**

Mostly yes, but only after narrowing some of the claims and tightening the interpretation of the experiments.

The paper provides convincing evidence for the following specific claims.

First, the proposal–repair architecture is effective in structured unknown-map MAPF. The strongest evidence is in Setting B, where local baselines degrade sharply in environments such as Mazes and Warehouses, while LENS remains much more stable. This directly supports the main practical claim of the paper that in partially observed, structurally constrained environments, learned local guidance plus explicit bounded repair is substantially more robust than purely reactive local coordination.

Second, the Neural Proposer is genuinely contributing. The ablations show substantial degradation when replacing the U-Net proposer with simple A* to subgoals or when removing ARG / cascade components. This supports the view that the architecture is truly compositional: coarse topology, local topo-metric inference, and local repair are all necessary.

Third, the local CBS-centered repair story is real. The profiling result that standard local CBS resolves 98.96% of conflict groups is important: it strongly suggests that the main repair mechanism emphasized in the paper is indeed what is driving performance, rather than the deeper fallback heuristics.

That said, several broader claims are only partially supported.

On originality, the paper makes a meaningful contribution, but the evidence does not support a broad claim around being the first to combine learning and search for MAPF. Prior work already includes:
- earned local graph-based MAPF policies under local observability, [1]
- deep learning-guided explicit MAPF search / LNS,  [2]
- and other hybrid planning-learning interfaces, including representation-shaping approaches for fixed planners. [3]

So the evidence supports a narrower originality claim: LENS contributes a particular learned-proposal / bounded-repair interface for centrally coordinated partially observable MAPF.

On the experiments, the Setting A vs Setting B split is a real strength, because it partly disentangles proposer validation from full-system evaluation. However, Setting A is best read as an architecture/proposer validation setup, since it is asymmetric in information and mixes different coordination mechanisms. Setting B is the core empirical contribution, but it necessarily uses a reduced baseline set because some strong methods are not portable to the unknown-map regime without redesign. Thus, the experiments support strong claims relative to the baselines portable to that regime, but not necessarily broader claims against all learned or search hybrids.

On robustness, the small degradation from Setting A to Setting B is suggestive and encouraging, but it does not by itself fully identify the source of robustness. The observed stability may reflect generalized local-topology inference in the proposer, the synchronized shared-map scaffold, the ARG prior, the explicit repair stack, or the interaction of all of these.

On efficiency, the paper does provide useful evidence that the core decision stack is relatively lightweight, since appendix profiling suggests that proposer with local CBS account for a small fraction of total time in large settings. However, this also implies that reported end-to-end runtime partly reflects the centralized synchronization / shared-map maintenance scaffold. Consequently, the narrower supported claim is that the decision stack itself appears efficient, while full system latency in large regimes is dominated by the centralized collaborative setting used in the experiments

Finally, in Setting B, SoC and runtime are computed over successful runs only. This is a reasonable convention, but it means that SoC should be interpreted jointly with SR, especially in difficult regimes where failures are frequent.

Overall, I think the submission provides accurate, clear, and largely convincing evidence for a meaningful architectural contribution, but some of the broader framing should be tightened so that the claims align exactly with what is demonstrated.

References:\
[1] van Knippenberg, Marijn, Mike Holenderski, and Vlado Menkovski. "Time-constrained multi-agent path finding in non-lattice graphs with deep reinforcement learning." Asian Conference on Machine Learning. PMLR, 2021.\
[2] Yan, Zhongxia, and Cathy Wu. "Neural neighborhood search for multi-agent path finding." The Twelfth International Conference on Learning Representations. 2024.\
[3] Captari, Marius, and Herke van Hoof. "Learned Representations Enhance Multi Agent Path Planning." ICML 2025 Workshop on Programmatic Representations for Agent Learning.

**Requested Changes:**

Critical:
1. Narrow the novelty claim.
The current framing around learning plus search for MAPF is too broad relative to prior work on learned local graph-based MAPF policies, deep learning-guided MAPF search, and other hybrid planning-learning interfaces (e.g., [1-3]).  The core novelty is more specific: learning a local topological proposal and invoking bounded repair only where predicted interaction structure requires it, in a centrally coordinated partially observable setting.

2. Make the setting explicit.
The method assumes centralized collaborative PO-MAPF with synchronized shared mapping, reliable communication, and centralized repair. This materially simplifies the decentralized problem and should be stated much more prominently in the framing.

3. Temper the critique of learned baselines.
The paper leans too much on generic “myopia/reactivity” as the explanation for learned-method failure. Recent work suggests that some failures in dense MAPF instead arise from weak interaction inductive biases such as pairwise bottlenecks / attention dilution (e.g., [4]).   Explicit repair is a compelling answer, but not obviously the only one.

4. Tighten the experimental interpretation.
Setting A is best read as proposer/architecture validation, not a full like-for-like systems comparison; Setting B uses a reduced comparator set because some strong methods are not portable to the unknown-map regime without redesign; and SoC/runtime in Setting B are conditioned on successful runs.  The paper should also be more cautious in interpreting the small A→B degradation, since the current experiments do not isolate whether robustness comes primarily from the proposer, the synchronized shared-map scaffold, or the repair stack.

References (cont):\
[4] Jain, Rishabh, et al. "Pairwise is Not Enough: Hypergraph Neural Networks for Multi-Agent Pathfinding." arXiv preprint arXiv:2602.06733 (2026).

---

> ### Author Response · Authors · 2026-04-18
>
> We thank the reviewer for the thorough and constructive assessment, particularly the observation that our contribution is best positioned as "a specific learned-proposal / bounded-repair interface" rather than a broad learning+search claim. We adopt this framing throughout the revision. The profiling evidence that standard L-CBS resolves 98.96% of conflict groups, which the reviewer highlighted, further supports this positioning. We address all four critical changes below; modifications in the paper are marked in Red color.
>
> **Q1: Narrow the novelty claim and make the centralized setting explicit [Critical #1, #2].**
>
> **R1:** We agree with both points and have revised the paper accordingly.
>
> (a) Setting explicitness.** The Abstract, Introduction, and Section 3 now explicitly state that LENS targets *Centralized Collaborative* PO-MAPF with a synchronized shared map ($\mathcal{M}_{shared}$) and centralized repair. Section 3 formalizes this as an MPOMDP and states that this abstraction materially simplifies the fully decentralized problem by factoring out SLAM and asynchronous map merging, isolating topology-aware conflict resolution as the studied challenge.
>
> **(b) Novelty scope.** Contributions 1 and 2 ( In Introduction) now frame the specific interface: a continuous local topo-metric estimator coupled with bounded discrete repair triggered only where the predicted interaction structure requires it. Section 2.4 adds a new paragraph positioning LENS against learned local graph-routing [van Knippenberg et al., 2021], deep learning-guided global search [Yan & Wu, 2024], and learned representations for fixed planners [Captari & van Hoof, 2025], clarifying that LENS differs by keeping combinatorial legality in L-CBS rather than in the global search layer or learned representations.
>
> **Q2: Temper the critique of learned baselines [Critical #3].**
>
> **R2:** We agree that framing learned-baseline failures purely as "myopia" is too coarse. Section 2.1 now cites Jain et al. [2026] and acknowledges that standard pairwise attention can suffer from attention dilution at high-order bottlenecks. Section 5.3 ("The Cost of Myopia") adds a clarifying note: the collapse of CS-PIBT in Maze and Warehouse reflects *both* 1-step myopia *and* limited interaction inductive biases. LENS is complementary to representation-level remedies (e.g., hypergraph encoders): it structurally bypasses the representational burden by delegating combinatorial interaction legality to L-CBS, rather than attempting to learn it.
>
> **Q3: Tighten experimental interpretation [Critical #4].**
>
> **R3:** We agree and have narrowed three distinct claims:
>
> **(a) Setting A is architecture validation, not like-for-like.** Section 5.2 now opens with an explicit note that Setting A is asymmetric (baselines receive oracle cost-to-go; LENS performs learned inference). It validates the U-Net as a heuristic approximator rather than establishing a system-level ranking.
>
> **(b) Setting B uses a reduced baseline set.** We preserved Appendix E.2's "Excluded Methods" subsection and added a sentence in Section 5.3 stating that Setting B claims hold against baselines portable to the unknown-map regime (SCRIMP, adapted LNS2-RL, adapted CS-PIBT), not against all hybrid methods.
>
> **(c) A→B degradation is entangled.** Section 5.3 now explicitly states that the minimal degradation is a *joint* product of U-Net generalization, the shared-map scaffold, the ARG prior, and the repair stack. While Table 4 ablates each component, fully isolating the U-Net's intrinsic robustness from system-level safeguards remains beyond the current experiments.
>
> **(d) SoC/runtime survivorship bias.** Section 5.1's Evaluation Metrics block now notes that Makespan and SoC are computed strictly over fully successful instances; Figure 2, Figure 3, and paragraph “Results Analysis” repeat this caveat.
>
> **Q4: Narrow the efficiency claim and interpret SoC jointly with SR [Reviewer's interpretation remarks].**
>
> **R4:** We agree both points are narrower than our original framing:
>
> **Efficiency.** Section 5.2 (Efficiency at Scale) now states that the core decision stack (U-Net + L-CBS) is lightweight, while end-to-end runtime in massive regimes is dominated by centralized state synchronization overhead. Appendix D (Table 9) provides the supporting latency profile on City-256: Map Update consumes 70.11% of total time, ARG planning 24.08%, while U-Net inference and L-CBS together account for only 4.58%. This confirms that observed runtime is primarily a property of the centralized collaborative setting, not of the hybrid architecture itself.
>
> **Metrics.** Section 5.1 explicitly states that Makespan and SoC are computed over fully successful instances; the updated Figure/Table captions reinforce that SoC should be read jointly with SR, especially in low-SR regimes where survivorship bias is largest.

---

### Review · Reviewer_YnYr · 2026-04-08

**Summary Of Contributions:**

This paper studies the MAPF problem with partial observability and proposes a framework named LENS that uses neural networks to learn and construct local navigation guidance (potential fields) and a search-based module to resolve conflicts.
Strengths: LENS works in an unknown environment; This framework uses less data compared to MAPF-GPT.

**Audience:**

Yes

**Audience Explanation:**

I believe this is an interesting work that uses ML to solve the MAPF problem from a different perspective.

**Broader Impact Concerns:**

None.

**Claims And Evidence:**

No

**Claims Explanation:**

The overall results of the paper are acceptable, and the experimental results are fine.

However, there are a few claims that may need clarification or support.

- Section 4.2, L-CBS
"Running concurrently on disjoint conflict groups, it mathematically guarantees collision-free execution for the n-step window."
May I ask if there is any proof or evidence to support this claim? I didn't find this in the paper.

- Section 4.3, Optimistic Assumption under Uncertainty,
"LENS mitigates this structural vulnerability through a tight feedback loop with the DEZD." Could you clarify which part of the experiment results supports this claim? Or is there any justification for this claim?

**Requested Changes:**

Major issues:

- Missing definitions of notations and concepts

When I was reading through the paper, I noticed that many notations and concepts were not defined. I did find the table of notations in the appendix. It explained some of the undefined notations, but it would be better if you could define them in the main text.

1. Up to Section 3 Problem Formulation, the paper assumes the audience knows about MAPF, and the problem definition section does not give a clear definition of the MAPF problem. For an audience without a MAPF background, this paper might be difficult to understand. For example, MAPF conflicts are discussed throughout the paper, but it is not clearly stated what the conflicts are in the problem definition. I would highly recommend clearly defining the MAPF concepts, including vertex and edge conflicts.

2. First paragraph of Section 3, i is not defined for O^i_t and u^i_t. I can see it may refer to the i-th agent, but it is better to define it explicitly. Global Guidance: M hat^i_t and g^i are not defined.

3. Section 4, Action Extraction, up to the current section, s^i_t is not defined. N() is not defined either in the text.

- Some claims need support or justification

1. Section 4.2, L-CBS
"Running concurrently on disjoint conflict groups, it mathematically guarantees collision-free execution for the n-step window."
May I ask if there is any proof or evidence to support this claim? I didn't find this in the paper.

2. Section 4.3, Optimistic Assumption under Uncertainty,
"LENS mitigates this structural vulnerability through a tight feedback loop with the DEZD." Could you clarify which part of the experiment results supports this claim? Or is there any justification for this claim?

3. (less important) Section 4.2, DEZD, is there a justification for choosing these values for intervention?

- The writing feels a bit "list-heavy" (less important)

Many paragraphs were written as lists of items, which is a little unusual to me.
For example, Section 4.3, second paragraph, the paragraph starts with a sentence, then it starts to list "Global Instance: ...", "Dynamic Connectivity: ...", "Optimistic Assumption under Uncertainty:...". The capitalization suggests they are like a subsubsection. Sometimes these items are numbered and sometimes they are not, e.g., Section 4.1 "1. Preservation of Spatial Topology:...".
It would be great if you could smooth the text a little bit.

Smaller issues:

- "In Mazes (Figure 2a), LENS achieves a 97.6% Success Rate at 32 agents and 78.7% at 64 agents. Remarkably, LENS achieves this competitive performance matching the centralized oracle LaCAM": the word "matching" might be too strong here since the success rate of LENS drops to 78.7% at 64 agents while LaCAM stays at near 100%.

- Double ".." at the end of the 1st paragraph of Section 4.2.

- Some mistakes in the references, please double check.
For example,

Runzhe Liang, Rishi Veerapaneni, Daniel Harabor, Jiaoyang Li, and Maxim Likhachev. Real-time lacam.
arXiv preprint arXiv:2504.06091, 2025. Title should be "Real-Time LaCAM for Real-Time MAPF".

Research Question (would be great if you could offer some insights in the paper):

How would unseen obstacles affect the learned approximation of the global potential field?
The obstacles (I assume these are the obstacles observed by agents' local views) are used as part of the 4-channel local spatial tensor. So would the proportion of unseen obstacles have some negative impact (e.g., misleading prediction) on the learned potential field? If it does have a negative impact, do you think encouraging some exploration of the environment would help to address it?

---

> ### Author Response · Authors · 2026-04-18
>
> We thank the reviewer for recognizing the relevance of LENS to the partially-observable MAPF setting and its data efficiency compared to MAPF-GPT. We address all five points below; all paper modifications are marked in Red color.
>
> **Q1: Missing definitions of notations and concepts (MAPF conflicts, $i$, $\hat{M}^i_t$, $g^i$, $s^i_t$, $\mathcal{N}(.)$).**
>
> **R1:** We agree. Section 3 has been restructured to formally define MAPF Vertex and Edge Conflicts. All agent-indexed notations ($i$, $s_t^i$, $g^i$, $o_t^i$, $u_t^i$) are now defined at first appearance in Section 3. The original $\hat{M}t^i$ has been unified as the shared map $\mathcal{M}_{shared}$. We also added an inline definition of $\mathcal{N}(\cdot)$ (4-connected neighbors) at its first use in Section 4.1.
>
> **Q2:  Some claims need support or justification: Proof or evidence for L-CBS safety guarantee.**
>
> **R2:** The guarantee rests on three pillars, now detailed in Section 4.2 (L-CBS paragraph) with citations to Sharon et al. (2015) and Boyarski et al. (2015):
>
> **(i) Spatio-Temporal Decoupling.** The server builds a Spatio-Temporal Conflict Graph $\mathcal{G}_{conf}$ over neural proposals and extracts its Connected Components as disjoint conflict groups; agents across groups share no spatio-temporal overlap within the $k$-step window.
>
> **(ii) Bounded Completeness within each group.** L-CBS uses Space-Time A* with explicit $(n_i, v, t)$ vertex/edge constraints. Standard CBS properties yield completeness and optimality within the $k$-step temporal bound.
>
> **(iii) Receding-horizon safety buffer.** Planning for $k$ steps (terminal "Wait") while executing only the first $n<k$ steps gives a temporal margin before the next centralized re-evaluation. Thus the executed $n$-step trajectories are strictly free of vertex and edge collisions *within the bounded execution window*—we have accordingly replaced the absolute "mathematically guarantees" wording with "bounding local execution safety" in Section 4.2.
>
> **Q3: • Some claims need support or justification: Justification for DEZD mitigation claim and intervention values.**
>
> **R3: Mitigation Claim:** The "structural vulnerability" arises strictly from the ARG's optimistic assumption: treating unknown areas as traversable inherently risks guiding agents into deep, unmapped dead-ends, causing local neural proposals to stagnate. We mitigate this via the DEZD feedback loop. We added an ablation (Table 4, "City, 128 agents"): disabling DEZD drops the Success Rate from 77.1% to 71.1% and surges the SoC by nearly 1000 steps. Agents without DEZD waste critical steps oscillating in these  unmapped dead-ends, confirming that DEZD effectively intercepts deadlock stagnation.
>
> **Intervention Values:** We added a note in Section 4.2 that the RHC horizon $k \in \{[5, 10]\}$ and stagnation threshold $\tau_{unique} \in \{[2, 6]\}$ were determined via grid search on a validation set. The chosen values ($k=10, \tau_{unique}=3$) strictly balance false positives (invoking expensive search for temporary dynamic congestion) and false negatives (wasting steps idling in hard topological deadlocks).
>
> **Q4: Writing style ("list-heavy", "matching" is too strong, typos, references).**
>
> **R4:** We appreciate the proofreading.
>
> (a) List-heavy structure. Section 4.1 (U-Net design rationale) and Section 4.3 (ARG properties) have been rewritten as continuous prose without bold labels or numbered items.
> (b) "Matching" softened. Section 5.2 now reads "maintains competitive robustness relative to the centralized oracle LaCAM."
> (c) Typos. The stray double period at the end of Section 4.2's first paragraph has been removed.
> (d) Reference. The citation for Liang et al. is corrected to "Real-Time LaCAM for Real-Time MAPF" (SoCS 2025).
>
> **Q5: Research Question on Unseen Obstacles.**
>
> **R5:** Thank you for the incisive question. We added Appendix G.4 (Figure 6) reporting an isolated study of U-Net prediction error versus the proportion of unseen obstacles on Cities-tiles maps with 2 agents. The regression shows a positive correlation between unseen-obstacle ratio and the normalized MSE of the predicted local potential field against the ground-truth BFS field, validating that the ARG's optimistic treatment of unknown cells induces heuristic distortion when critical structure lies outside the FOV. Currently LENS handles this reactively via the Defense Cascade; we agree that proactive remedies (uncertainty-aware U-Net inputs, entropy-reduction objectives) are a natural next step and have listed this explicitly under Future Work.

---

### Review · Reviewer_4GUh · 2026-04-10

**Summary Of Contributions:**

The paper studies a multi-agent navigation / coordination problem in a partially observed grid-like environment with hidden structure and limited local observations. The idea is to combine local reactive guidance (learned) with a higher-level coordination mechanism (search) so that agents can move safely and efficiently despite incomplete information. The method introduces several components intended to support local decision-making, agent interaction, and overall coordination under uncertainty. Empirically, the paper aims to show that the proposed design improves performance relative to existing approaches in a variety of settings, including unknown environments that require online collaborative mapping.

The paper presents three claims: two are about the effects of particular structural components, and a third is about data-efficiency and online robustness. The experiments primarily support the claims in the third claimed contribution (which is the only contribution that describes something measurable). Ablations illustrate the effects of removing some of the components of the method. The claimed contributions are not yet clearly differentiated from past work, the problem setup is not formulated precisely enough to support the method description, and the presentation of the approach is difficult to follow because the intended role of each component remains unclear.

**Audience:**

Yes

**Audience Explanation:**

The method, although hard to understand how it works and in what settings it applies, appears to perform well and has a straightforward inspiration of blending learning for local prediction and search for global coordination.

**Claims And Evidence:**

No

**Claims Explanation:**

Some of the claims are unclear:

- Contribution 1: Unclear what differentiates this contribution from past work and this claim is not made measurable (or it's not obvious what evidence supports it). Needs to be rephrased to make it clearer what's new. It would help if this claim were made measurable, e.g. if "mitigating reactive myopia" was shown experimentally and measured.
- Contribution 2: Citation(s) needed for "passive shielding paradigm", this is unclear. This claim needs to be made measurable (or obviously connected to evidence that supports it), specifically, the claim appears to be that the approach "guarantees the optimal resolution of structured deadlocks within localized spatio-temporal windows while strictly maintaining real-time execution frequencies".

Issues with problem formulation and method description:
- The problem formulation is fairly underspecified and hard to follow. It's missing mathematical formalization of the hidden grid graph and of the problem objective. The relationship between the observations of each of the agents is unclear. The observation space of each of the agents is also unclear (it reads as if it's an 11x11 occupancy map, but it's unclear). Furthermore, the relationship between the problem setting and existing MARL and/or POMDP settings is unclear. I strongly suggest using POMDPs to describe the problem setting (and I believe the local potential fields would also be more clearly described as approximate local value functions of some near-optimal policy, although I'm not sure).

Other issues:
- Missing self-contained definitions of the evaluation metrics.
- Visualizations of the inputs and outputs of each of the components described in the method flowchart would help in explaining the method.
- Fig. 1 is hard to follow overall -- the separation between training and testing is unclear, the types of the inputs and outputs of each of the components are unclear.

**Requested Changes:**

As mentioned elsewhere, please address weaknesses regarding the clarity of the claims, making the problem setting clear and precise, and connecting the proposed method and insights regarding each of its components to the precise formulation of the problem.

---

> ### Author Response · Authors · 2026-04-18
>
> We thank the reviewer for the constructive feedback. The critiques have reshaped Sections 3, 4, and 5.1 of the revision. Paper modifications are in red.
>
> **Q1: Contribution 1 is not measurable and insufficiently differentiated from prior work. Contribution 2 contains an overly strong unsupported guarantee and lacks citations for "passive shielding paradigm."**
>
> **R1:**
>
> **Contribution 1 — measurable horizon claim + differentiation.** We rewrote Contribution 1 (Introduction) to contrast LENS explicitly with both 1-step reactive policies (CS-PIBT) and conventional hybrid frameworks (LNS2-RL), which invert the learn/search roles relative to ours along two axes: search is invoked locally over anticipated conflict groups rather than globally over the neighborhood, and learning produces a continuous potential field rather than discrete repair actions. The myopia mitigation is now made measurable via a new **Long Corridor Case Study (Appendix B, Table 7, Figure 4)**: at 16 agents on corridor-dominant maps, CS-PIBT collapses to 20.0% SR, whereas LENS maintains 56.2% SR under identical map information, isolating the contribution of extended horizon from map-awareness. Maze-64 results in Figure 2/3 corroborate this in dense structured regimes.
>
> **Contribution 2 — "reactive shielding paradigm" + bounded guarantee.** Corrected the terminology to "reactive shielding paradigm" and added citations. The earlier absolute claim has been softened to **"bounded collision-free coordination within localized spatio-temporal windows"**, supported in Section 4.2 (L-CBS) via three mechanisms: (i) spatio-temporal decoupling into disjoint conflict groups, (ii) bounded completeness of Space-Time A* under CBS constraint propagation, and (iii) a receding-horizon safety buffer ($n < k$). Empirically corroborated by the ablation in **Table 4** (removing L-CBS drops SR) and by cascade usage statistics in **Appendix D, Table 8** (L-CBS alone resolves 98.96% of conflict groups).
>
> **Q2: Issues: Problem formulation and method description, the relationship between the problem setting and existing MARL and/or POMDP settings is unclear.**
>
> **R2:**
> Section 3(setting) is fully reformulated as a Multi-Agent Partially Observable Markov Decision Process (MPOMDP) and explicitly defined all components of the tuple $\langle\mathcal{I},\mathcal{S},\mathcal{A},\mathcal{T},\Omega,\mathcal{O},\mathcal{R}\rangle$.
>
> 1. Relationship to MARL/POMDP: We clarified that unlike standard MARL which focuses on fully decentralized execution and communication learning, our setting is a Centralized Collaborative MPOMDP. Agents perform decentralized perception but utilize a synchronized edge server for shared mapping and centralized conflict resolution.
> 2. Observation Space: We formulated the observation space $\Omega$ as a strict $11 \times 11$ local FOV occupancy grid.
> 3. Local Potential Fields: We described the neural potential fields in Section 3 as approximating the optimal local value function (cost-to-go) to guide the agents.
>
> **Q3: lacks definitions of evaluation metrics; Fig. 1 is difficult to follow due to unclear training/testing separation; ambiguous input/output types for each component; insufficient visualizations to illustrate the method flow.**
>
> **R3:**
>
> **Metrics.** Section 5.1 now gives definitions of Success Rate (SR), Individual Success Rate (ISR), Makespan, and Sum-of-Costs (SoC), and notes that Makespan/SoC are computed only over fully successful instances to avoid survivorship bias.
>
> **Figure 1.** The flowchart is updated with colour-coded block types (blue = raw inputs $\Omega$ and guidance; grey = final collision-free actions $u_t$) and its caption clarifies that **Figure 1 depicts only the online inference pipeline**. Supervised training procedure is described in Section 4.1 "Data Generation" and its inputs/outputs are visualized in Appendix G.2 (Figure 5).
>
> **Component I/O visualization.** Appendix G.2 (Figure 5) visualizes the two spatial input channels (Obstacles, Other Agents), the goal direction vector, the global BFS potential field, and the cropped local Prediction Label used as the supervision target; the remaining two channels (Self-Position, Projected Local Target) share the same high-intensity encoding as the Other Agents channel and are described in the accompanying text.
>
> **Q4: Link method and insights with each components to the precise formulation of the problem.**
>
> **R4:**
> Section 4 added a paragraph that maps each LENS module to the MPOMDP tuple: the Adaptive Guidance Mechanism recovers the hidden graph $G$ and yields $\mathbf{v}_{guide}$; the Neural Proposer regresses the dense potential field $\hat{\Phi}$ from $o_t^i \in \Omega$ as an approximate local value function; the Defense Cascade enforces the transition constraints $\mathcal{T}$ (vertex/edge conflict avoidance) by bounded search over anticipated conflicts and produces $\mathbf{u}_t$. Figure 1 annotates the data flow with the same symbols.

---

### Comment · Action_Editor_sD2s · 2026-04-17
**Author-Reviewer Discussion Phase**

Hello Authors,

This is a reminder that we are currently in the Author–Reviewer discussion phase for this paper. Please use this period to respond to the reviewers’ comments and/or revise the manuscript as needed. After one week, reviewers will be able to submit their final decisions based on your rebuttal.

---

### Decision · Action_Editor_sD2s · 2026-05-15

**Recommendation:** Accept as is

**Additional Comments:**

The paper introduces LENS, a hybrid architecture that decouples local topological inference from multi-agent collision resolution in partially observable MAPF.  LENS learns a dense local potential field while invoking localized CBS-based repair over predicted conflict regions. Empirical evaluations suggest that this approach performs relatively better than existing approaches, specifically in unknown environments.


The authors have addressed most of the reviewer concerns through a strong and thorough rebuttal. I commend the authors on strong rebuttal and recommend to keep the revisions it in the final camera-ready version.

**Audience:**

Yes

**Audience Explanation:**

The paper addresses a relevant and challenging problem at the intersection of multi-agent learning, planning, and partially observable decision making. It will be of interest to MAPF and multi-agent community at TMLR.

**Claims And Evidence:**

Yes

**Claims Explanation:**

All the three reviewers recommended accept and confirm the claims are accurate.

The reviewers initially raised concerns around the contribution claims during review. These concerns were addressed by the authors and substantially modified. Overly strong claims were softened and corroborating empirical evidence was added during rebuttal.